# EFFICIENT CALIBRATION AS A BINARY TOP-VERSUS-ALL PROBLEM FOR CLASSIFIERS WITH MANY CLASSES

## ABSTRACT

Most classifiers based on deep neural networks associate their class prediction with a confidence, usually defined as the maximum predicted output value. This value is often a by-product of the learning step and may not correctly estimate the classification accuracy, which impacts real-world usage. To be reliably used, the confidence requires a post-processing calibration step. Data-driven methods have been proposed to calibrate the confidence of already-trained classifiers. Still, many methods fail when the number of classes is high and per-class calibration data is scarce. To deal with a large number of classes, we propose to reformulate the confidence calibration of multiclass classifiers as a single binary classification problem. Our *top-versus-all* reformulation allows the use of the binary cross-entropy loss for scaling calibration methods. Contrary to the standard one-versus-all reformulation, it also allows the application of binary calibration methods to multiclass classifiers with efficient use of scarce per-class calibration data and without degradation of the accuracy. Additionally, we solve the problem of scaling methods overfitting the calibration set by introducing a regularization loss term during optimization. We evaluate our approach on an extensive list of deep networks and standard image classification datasets (CIFAR-10, CIFAR-100, and ImageNet). We show that it significantly improves the performance of existing calibration methods. Code to replicate some of the experiments can be consulted at `https://anonymous.4open.science/r/top-versus-all-calibration-6898`.

## 1 INTRODUCTION

The huge performance increase of modern deep neural networks (DNN) and their potential deployment in real-world applications (medical, transportation, or military) has made the question of reliably estimating the probability of wrong decisions a key concern. When such components are expected to be embedded in safety-critical systems, estimating this probability is crucial to mitigate catastrophic behavior.

One way to address this issue is to solve it as an uncertainty quantification problem (Abdar et al., 2021; Gawlikowski et al., 2023), where the uncertainty value computed for each prediction is typically used either as a confidence to accept or reject the decision proposed by the DNN for selective classification (Geifman & El-Yaniv, 2017) or out-of-distribution detection (Hendrycks & Gimpel, 2017), or as a measure to control active learning (Li & Sethi, 2006) or reinforcement learning based systems (Zhao et al., 2019).

Uncertainty quantification can also be wrong: a common way to assess the quality of uncertainty values is to measure their ability to predict the true probability of a correct decision, i.e., their accuracy. In this case, a predictive system is said to be *calibrated*, which means that taking a decision according to its uncertainty will induce an accuracy with the same value: uncertainty values can then be used as a reliable control of decision-making.

We are interested in producing an uncertainty indicator for decision problems where the input is high dimensional and the decision space large, typically image classifiers with hundreds or thousands of classes. For this kind of problem, DNNs are common predictors, and their outputs can be used to provide at no cost an uncertainty value – i.e., without necessitating heavy estimation such as Bayesian sampling or ensemble methods. Indeed, most neural architectures for classification

instantiate their decision as a soft-max layer, where the maximum value can be interpreted as the maximum of the posterior probability and, therefore, as a confidence.

Unfortunately, uncertainty values computed in this way are often miscalibrated. DNNs have been shown to be over-confident (Guo et al., 2017), meaning their confidence is higher than their accuracy: predictions with 90% confidence might be correct only 80% of the time. A later study (Minderer et al., 2021) suggests that model architecture impacts calibration more than model size, pre-training, and accuracy.

These studies show that it is difficult to anticipate the calibration level of confidences computed directly from DNNs, and argue for a complementary post-processing calibration. This calibration process can be seen as a learning step that exploits data from a calibration set, distinct from the training set, and is used to learn a function that remaps classifier outputs into better-calibrated values. This process is typically lightweight and decoupled from the issue of improving model performance. A standard baseline for post-processing calibration is temperature scaling (Guo et al., 2017), where the penultimate logit layer is scaled by a coefficient optimized on the calibration set.

Many post-processing calibration methods have been developed for binary classification models (Zadrozny & Elkan, 2001; 2002; Platt, 1999). Applying them to multiclass classifiers requires some adaptation. One standard approach reformulates the multiclass setting into many one-versus-all binary problems (one per class) (Zadrozny & Elkan, 2002). One limitation of this approach is that it does not scale well. When the number of classes is large, the calibration data is divided into highly unbalanced subsets that do not contain enough positive examples to solve the one-versus-all binary problems.

The main idea of our work is to reformulate the multiclass setting into a *single* binary problem. It can be phrased as: "Is the prediction correct?". In this new setting, the prediction becomes a scalar value: the confidence (which is defined as the maximum class probability), and the label becomes binary: 1 if the predicted class was correct, 0 otherwise. The objective is that the confidence accurately describes whether the prediction was correct, regardless of the class. We show that this novel approach, which we call *top-versus-all* (TvA), significantly improves the performance of standard calibration functions: temperature and vector scaling (Guo et al., 2017), Dirichlet calibration (Kull et al., 2019), histogram binning (Zadrozny & Elkan, 2001), isotonic regression (Zadrozny & Elkan, 2002), Beta Calibration (Kull et al., 2017), and Bayesian Binning into Quantiles (Naeini et al., 2015). We also introduce a simple regularization for vector or Dirichlet scaling that mitigates overfitting when the number of classes is high relative to the calibration data size. We demonstrate the approach on several image classification datasets: CIFAR-10, CIFAR-100, and ImageNet, with many different modern pre-trained models.

Our main contributions are:

- We develop the top-versus-all approach to the confidence calibration of multiclass classifiers, transforming the problem into a single binary classification. This setting significantly improves the performance of scaling methods (such as temperature scaling, vector scaling, and Dirichlet calibration). It also allows binary methods (such as histogram binning, isotonic regression, and beta calibration) to efficiently use scarce per-class calibration data and preserve the classifier's accuracy. It can be applied to many existing calibration functions.

- We introduce a simple regularization, allowing the competitive performance of vector scaling and Dirichlet calibration when the number of classes is high.

- We demonstrate the scalability of our approach by conducting extensive experiments with state-of-the-art image classification models on CIFAR-10, CIFAR-100, and ImageNet.

## 2 RELATED WORK

**Calibration**  Different variations of the calibration problem exist. One can consider confidence (Guo et al., 2017), class-wise (Kull et al., 2017), top-$r$ (Gupta et al., 2021), top-label (Gupta & Ramdas, 2022), or strong (Vaicenavicius et al., 2019; Widmann et al., 2019) calibration. In this work, we tackle the problem of confidence calibration. This problem is less difficult and demanding than the others because it only considers the confidence whereas other problems consider up to the

full probability vector. Still, confidence calibration is enough for many practical use cases. For recent surveys on calibration, we refer to Filho et al. (2023); Wang (2023).

**Training calibrated networks**   A family of approaches aims to train a neural network in specific ways to improve their calibration. They generally design a loss term to train better-calibrated networks Kumar et al. (2018); Thulasidasan et al. (2019); Cheng & Vasconcelos (2022); Karandikar et al. (2021). While these methods can directly optimize calibration during the training phase of the neural networks, they require a high development time and often compromise accuracy.

**Post-processing (or post hoc) calibration**   Another family of approaches aims to calibrate already-trained networks. This lowers the development time and decouples the accuracy optimization and the calibration. Some methods optimize some parameters to scale the logits. They include Platt scaling (Platt, 1999), temperature and vector scaling (Guo et al., 2017). Kull et al. (2019); Zhang et al. (2020) developed ensemble temperature scaling. Scaling can be combined with binning (Kumar et al., 2019). Instead of using the network logits or probabilities, its features can also be leveraged (Lin et al., 2022). Gaussian processes can be used but are more computationally heavy during inference (Wenger et al., 2020). Large vision-language models can also be calibrated (LeVine et al., 2023). Many methods were developed for the binary classification setting. They include histogram binning (Zadrozny & Elkan, 2001), isotonic regression (Zadrozny & Elkan, 2002), Bayesian binning with quantiles (Naeini et al., 2015), and beta calibration (Kull et al., 2017). We include more details on the methods we used in the paper in Section 3.1

**Scaling methods for calibration**   Temperature scaling (Guo et al., 2017) is a popular post-processing calibration method derived from Platt scaling (Platt, 1999). The logits vector $z$ is scaled by a single temperature coefficient $T$. The scaled logits vector then passes through the $\sigma$ softmax layer, and the temperature value affects the resulting probabilities as $\sigma(z/T)$. A temperature greater than 1 makes the probability vector more uniform, reducing the overconfidence of the networks, while a temperature less than 1 reduces their underconfidence. Since this scaling does not change the ranking of the logits and probabilities, the class prediction is unchanged when applying the decision after calibration. Because of its simplicity and good performance, temperature scaling is the default baseline for post-processing calibration. Vector scaling (Guo et al., 2017) is another multi-class extension of Platt scaling. This time, the logits vector is multiplied element-wise by another vector $v \in \mathbb{R}^L$: $\sigma(z \circ v)$, where $\circ$ is the Hadamard product. Vector scaling is more expressive than temperature scaling because each class logit has a different scaling coefficient. Since scaling the classes differently can change the ranking of the class probabilities, it is possible that the class prediction can change. Several works show that vector scaling has good performance in many cases (Guo et al., 2017; Nixon et al., 2019; Kull et al., 2019). Matrix scaling can also be considered for additional expressiveness but is difficult to apply without overfitting (Guo et al., 2017). Dirichlet calibration (Kull et al., 2019) proposes a regularization strategy for matrix scaling.

**Binary methods for calibration**   Many methods have been developed for the binary setting. Histogram Binning (Zadrozny & Elkan, 2001) divides the prediction into $B$ bins according to the predicted probability. For each bin, a calibrated probability is computed from the calibration data. The probability becomes discrete: it can only take $B$ values. The method usually follows the one-versus-all approach of multiclass models by learning a different histogram for each class. Some modifications can make it outperform scaling methods (Gupta & Ramdas, 2022; Patel et al., 2020). Isotonic regression (Zadrozny & Elkan, 2002) is a generalization of histogram binning that learns a piecewise constant function to remap probabilities. Bayesian binning with quantiles (Naeini et al., 2015) brings Bayesian model averaging to histogram binning. Beta calibration (Kull et al., 2017) uses a beta distribution to obtain a calibration mapping. Venn-Abers predictors (Vovk & Petej, 2012) apply to binary classifiers and are always well calibrated.

**Multiclass to Binary**   Using binary calibration methods for a multiclass classifier requires adapting the multiclass setting. This is usually done with a one-versus-all approach (Zadrozny & Elkan, 2002; Guo et al., 2017). The multiclass setting is decomposed into L one-versus-all problems: one binary problem for each class. L calibrators are derived, each one independently calibrating the probability of one class. One problem of this approach is the lack of calibration data for each of the L problems for many classes (if we take 25000 ImageNet samples for calibration, each of the 1000

binary calibration problems has only 25 images available). Another issue is that because each component of the probability vector is changed independently, the model prediction may change. Gupta & Ramdas (2022) introduce the notion of top-label calibration, i.e. confidence calibration with an additional conditioning on the predicted class (top-label). They describe a general multiclass-to-binary to develop top-label calibrators. In their framework, one calibrator is learned for each *predicted* class so it suffers from the same issues as the one-versus-all framework described above. Patel et al. (2020); Zhang et al. (2020) merge the classwise calibration sets into a single one. This is similar to TvA, except our approach only considers the confidence, as other probabilities are unimportant for confidence calibration. On the opposite side, Cheng & Vasconcelos (2022) derive $L(L-1)/2$ pairwise binary problems. The approach requires training the classifier from scratch, and its performance is negatively affected by a growing number of classes (only tested up to 100).

## 3 Top-versus-all approach to confidence calibration

### 3.1 Problem setting

**Confidence calibration of a classifier** We consider the image classification problem where an input image $x$ is associated with a class label $y \in \mathcal{Y} = \{1, 2, ..., L\}$. Let us define a neural network image classifier $f(x)$ where the last layer is a softmax. The softmax function $\sigma$ transforms logits $z$ of the neural network into probabilities $\sigma(z) = f(x)$. The classifier prediction is the most probable class $\hat{y} = \arg\max_{j \in \mathcal{Y}} f_j(x)$ with $f_j(x)$ referring to the probability of class $j$, and the confidence is $\hat{p} = \max_{j \in \mathcal{Y}} f_j(x)$. Note that the confidence is the maximum class probability, and we use these terms interchangeably. With $y$ the real label, we consider the classical *confidence* calibration definition (Guo et al., 2017) that says that the classifier $f$ is calibrated on a given data distribution if:

$$P(\hat{y} = y | \hat{p} = p) = p, \quad \forall p \in [0, 1] \tag{1}$$

where the probability is over the data distribution. It means the probability of being correct when the confidence is around $p$ is indeed $p$. For instance, if we consider all predictions with a confidence of $90\%$, they should be correct $90\%$ of the time. Because 1 cannot be computed from a finite number of samples, empirical approximations are required. The Expected Calibration Error 2 is a way to compute the calibration error. We focus on confidence calibration and not the calibration of the full probability vector. This is because calibrating the full probability vector is much more difficult, especially when the number of classes is high, and is useless for many applications. Indeed, many applications only use confidence values, such as selective classification (Geifman & El-Yaniv, 2017), out-of-distribution detection (Hendrycks & Gimpel, 2017), or active learning (Li & Sethi, 2006).

**Post-processing calibration** We are interested in the case where a classifier has already been trained, and the goal is to improve its calibration, i.e., we address post-processing calibration. Post-processing calibration methods aim to remap the classifier probabilities to better-calibrated values without modifying the classifier. They typically use a calibration set different from the training set to optimize parameters or learn a function. We focus on post-processing calibration because it allows better use of off-the-shelf models, and it decouples the model training (optimizing for accuracy) and calibration. Both of these advantages significantly lower the development cost to obtain a well-performing and well-calibrated model in opposition to optimizing calibration at training time.

**Metrics** To quantify the calibration errors, several metrics exist. The most common is the Expected Calibration Error (ECE) (Naeini et al., 2015). It approximates the calibration error by partitioning the predictions into $B$ bins according to the confidence. The absolute difference between the accuracy and confidence is computed for each subset of data contained in the bins. The final value is a weighted sum of the differences of each bin.

$$\text{ECE} = \sum_{b=1}^{B} \frac{n_b}{N} |\text{acc}(b) - \text{conf}(b)| \tag{2}$$

where $n_b$ is the number of samples in bin $b$, $N$ is the total number of samples, $\text{acc}(b)$ is the accuracy in bin $b$, and $\text{conf}(b)$ is the average confidence in bin $b$. ECE can be interpreted visually by looking at diagrams in Figure 1: ECE computes the sum of the red bars (difference between bin accuracy and average confidence) ponderated by the proportion of samples in the bin. ECE has flaws: the

estimation quality is influenced by the binning scheme, and it is not a proper scoring rule (Gneiting & Raftery, 2007; Vaicenavicius et al., 2019; Nixon et al., 2019). Variants of ECE have been developed. For instance, Kull et al. (2019) define a classwise-ECE, Nixon et al. (2019); Minderer et al. (2021) use bins with equal mass (same number of samples per bin). Gupta & Ramdas (2022) defines top-label-ECE, similar to ECE but additionally conditioned on the predicted class. Despite its flaws, ECE remains the standard comparison metric and fits well with the confidence problem that we tackle.

## 3.2 TOP-VERSUS-ALL APPROACH

---

**Algorithm 1** Top-versus-all

---

**Input**:
$S_{cal}$: $\{(x_i, y_i)\}_{i=1}^N$ the calibration set
$f$: the classifier
$g$: the calibrator
**Pre-processing**:
$f^b \leftarrow \max_{j \in \mathcal{Y}} f_j$ ▷ Build binary classifier
$s_i \leftarrow f^b(x_i)$ ▷ Compute confidences
$\hat{y}_i \leftarrow \arg\max_{j \in \mathcal{Y}} f_j(x_i)$ ▷ Compute class predictions
$c_i \leftarrow \mathbf{1}_{\hat{y}_i = y_i}$ ▷ Compute predictions correctness
$S_{cal}^b \leftarrow \{(s_i, c_i)\}_{i=1}^N$ ▷ Build binary calibration set
**Learn calibrator**:
**if** $g$ **is** scaling method **then**
    loss $l := $ Binary Cross-Entropy
    **if** $g$ **is** vector or Dirichlet scaling **then**
        loss $l \leftarrow l + \lambda l_{reg}$ ▷ Add regularization
    **end if**
    Learn $g$ to calibrate $f^b$ by minimizing loss
**else if** $g$ **is** binary method **then**
    Learn $g$ to calibrate $f^b$ by following method
**end if**
**Inference**:
Use calibrator $g$ to calibrate confidences from $f^b$

---

**General presentation** We note that the definition 1 and the standard metrics ECE only consider whether the confidence reflects the probability of making an accurate prediction. The remaining probabilities are not taken into account. However, many calibration methods use all probabilities, not just confidence. We aim to simplify the process of confidence calibration by reformulating the problem of calibrating multiclass classifiers into a *single* binary problem. This problem can be formulated as: "Is the prediction correct?". In this setting, we transform the model prediction from a probability vector to a scalar: the confidence (the maximum class probability). The remaining probabilities are discarded. We transform the label from a class label (or one-hot encoding) to a binary number (1 or 0) representing whether the class prediction was correct. The goal is that the confidence accurately describes whether the prediction was correct, regardless of the class. More formally, the classification output for our binary problem becomes the confidence $s(x) = \max_{j \in \mathcal{Y}} f_j(x)$. The ground truth label becomes a binary representation of the prediction correctness: $y_b = \mathbf{1}_{\hat{y}=y}$ with $\hat{y} = \arg\max_{j \in \mathcal{Y}} f_j(x)$ and $\mathbf{1}$ the indicator function. Algorithm 1 recapitulates our approach.

With our approach, only the confidence needs to be calibrated, and it does so by efficiently using all the samples in the calibration set. The standard approaches, both for scaling methods and binary methods with one-versus-all, consider classes separately. Calibration data is divided between the classes, which becomes an issue for data sets with many classes.

In the following paragraphs, we explain how top-versus-all applies to scaling and binary methods.

**Top-versus-all for scaling methods** Now let us see how top-versus-all applies to scaling methods. The temperature $T$ and the vector $v$ are typically optimized to lower the cross-entropy loss on a calibration set separate from the training set. In our top-versus-all setting, the output becomes a scalar and the label binary. The natural loss for this formulation is the binary cross-entropy:

$$l_{BCE}(x,y) = -\big(y_b \cdot \log s(x) + (1 - y_b) \cdot \log(1 - s(x))\big) \tag{3}$$

Minimizing this loss results in confidence estimates that more accurately describe the probability of being correct, regardless of the $L-1$ less likely class predictions. Using the binary cross entropy as a calibration loss makes an important difference when compared to the usual multiclass cross entropy or negative log-likelihood (Guo et al., 2017). The multiclass cross-entropy loss takes into account the probability of the *correct* class, while the binary cross-entropy takes into account the probability of the *predicted* class (i.e., the confidence). When the predicted class is correct, minimizing both losses directly increases the confidence. A different behavior occurs when the prediction is wrong: minimizing the cross-entropy loss increases the probability of the correct class (thus only *indirectly* decreasing the confidence), but minimizing the binary cross-entropy loss *directly* decreases the confidence.

**Top-versus-all for binary methods** Methods such as histogram binning, isotonic regression, Bayesian Binning into Quantiles, and beta calibration originally apply to binary settings. Using

them in a multiclass setting is typically done with a one-versus-all approach, inappropriate for a high number of classes, as discussed in Section 2. Our reformulation transforms the multiclass setting into a *single* binary problem. This allows binary calibration methods to be applied using the full calibration data set (not just the per-class subsets). Another advantage is that since calibration methods now operate on confidence alone, the class prediction is already done. In contrast to the one-versus-all approach, the classifier's prediction and accuracy are unaffected.

**Regularization of scaling methods for a high number of parameters**   In addition to the top-versus-all reformulation, we present here another contribution. Many current works on calibration evaluate their approach on the CIFAR dataset, which contains 10 or 100 classes. After experimenting on more complex datasets such as ImageNet which contains $L = 1000$ classes, we found that vector scaling ($L$ parameters) and Dirichlet calibration ($L^2 + L$ parameters) overfit the calibration set. Overfitting can be reduced with a simple L2 regularization that penalizes when the vector coefficients are far from the reference value 1. We propose to add a complementary loss term for vector scaling and Dirichlet calibration:

$$l_{L2}(v) = \frac{1}{L} \sum_{i=1}^{L} (v_i - 1)^2 \qquad (4)$$

This regularization allows these methods to take advantage of their additional expressiveness without being subject to overfitting. The loss for vector scaling becomes $l(x, y, v) = l_{BCE}(x, y) + \lambda l_{L2}(v)$ where $\lambda$ is a hyperparameter. Dirichlet calibration uses this loss plus matrix regularization terms.

## 4 EXPERIMENTS

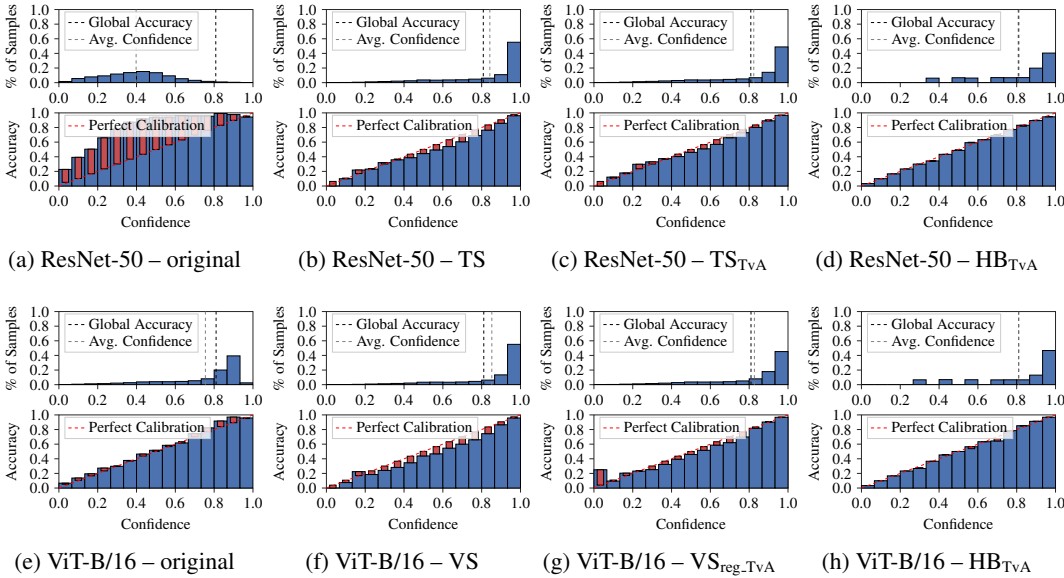

(a) ResNet-50 – original  (b) ResNet-50 – TS  (c) ResNet-50 – TS$_{\text{TvA}}$  (d) ResNet-50 – HB$_{\text{TvA}}$

(e) ViT-B/16 – original  (f) ViT-B/16 – VS  (g) ViT-B/16 – VS$_{\text{reg\_TvA}}$  (h) ViT-B/16 – HB$_{\text{TvA}}$

Figure 1: Reliability diagrams for ResNet-50 and ViT-B/16 when using temperature scaling (TS), vector scaling (VS), and histogram binning (HB) on ImageNet. The subscript $_{\text{TvA}}$ signifies that the TvA reformulation was used, and $_{\text{reg}}$ means our regularization 4 was applied. As the methods improve the calibration, the accuracy per bin will get closer to the true accuracy and the average confidence will get closer to the global accuracy.

**Datasets**   We experimented on three image classification datasets. CIFAR-10 and CIFAR-100 (Krizhevsky, 2009) contain 60000 32x32 images corresponding to 10 and 100 classes. Data is split into subsets of 45000/5000/10000 images for train/validation/test. We follow the standard practice of using the validation set as the calibration set and evaluate the results on the test set. ImageNet (Deng et al., 2009) contains 1.3 million images from 1000 classes. We randomly split the validation test of 50000 images into a calibration set and a test set for evaluation. We used a calibration set of size 25000 (with the test set containing the rest), following Guo et al. (2017).

Table 1: ECE in % (lower is better). The best method for a given model is in bold. Mean relative improvements from TvA are shown (negative values mean the ECE has been reduced). Methods in purple impact the model prediction, potentially degrading accuracy; methods in teal do not. Accuracies can be seen in Table 4.

(a) CIFAR-10

| Model | Uncal. | I-Max | TS | $TS_{TvA}$ | VS | $VS_{reg,TvA}$ | DC | $DC_{reg,TvA}$ | HB | $HB_{TvA}$ | Iso | $Iso_{TvA}$ | Beta | $Beta_{TvA}$ | BBQ | $BBQ_{TvA}$ |
|---|---|---|---|---|---|---|---|---|---|---|---|---|---|---|---|---|
| ResNet-50 | 1.82 | 1.28 | 1.12 | **0.78** | 0.98 | 0.92 | 0.98 | 0.93 | 1.06 | 1.25 | 1.31 | 1.28 | 2.40 | 1.33 | 0.93 | 1.34 |
| ResNet-110 | 2.56 | 0.56 | 1.30 | 1.36 | 1.28 | 1.34 | 1.28 | 1.34 | 0.96 | **0.34** | 1.30 | 0.69 | 2.83 | 1.79 | 1.23 | 1.20 |
| WRN | 1.25 | 0.34 | 0.95 | 0.95 | 1.02 | 1.11 | 1.02 | 1.11 | 0.58 | **0.28** | 0.99 | 0.45 | 1.49 | 1.07 | 0.68 | 0.38 |
| DenseNet | 1.53 | 0.77 | 1.29 | 1.46 | 1.23 | 1.42 | 1.20 | 1.41 | 1.04 | **0.62** | 1.16 | 0.69 | 2.15 | 1.92 | 1.05 | 1.05 |
| Mean improvement | | | -3% | | 6% | | 7% | | -35% | | **-36%** | | -30% | | -1% | |

(b) CIFAR-100

| Model | Uncal. | I-Max | TS | $TS_{TvA}$ | VS | $VS_{reg,TvA}$ | DC | $DC_{reg,TvA}$ | HB | $HB_{TvA}$ | Iso | $Iso_{TvA}$ | Beta | $Beta_{TvA}$ | BBQ | $BBQ_{TvA}$ |
|---|---|---|---|---|---|---|---|---|---|---|---|---|---|---|---|---|
| ResNet-50 | 6.52 | **0.82** | 4.83 | 2.63 | 5.30 | 2.30 | 5.29 | 2.31 | 8.60 | 1.34 | 5.44 | 0.99 | 5.33 | 3.48 | 9.14 | 2.65 |
| ResNet-110 | 7.88 | **1.23** | 4.68 | 3.97 | 5.25 | 3.00 | 5.20 | 3.03 | 9.85 | 1.39 | 6.04 | 1.68 | 5.63 | 4.57 | 7.83 | 1.67 |
| WRN | 4.31 | 0.83 | 4.22 | 2.81 | 4.26 | 2.00 | 4.26 | 1.96 | 9.67 | 1.11 | 4.52 | 0.92 | 4.32 | 2.91 | 9.88 | **0.72** |
| DenseNet | 5.17 | **1.03** | 4.05 | 2.17 | 4.35 | 2.00 | 4.31 | 1.98 | 9.09 | 1.12 | 4.52 | 1.19 | 4.71 | 2.58 | 9.92 | 1.38 |
| Mean improvement | | | -35% | | -52% | | -52% | | **-87%** | | -77% | | -33% | | -82% | |

(c) ImageNet

| Model | Uncal. | I-Max | TS | $TS_{TvA}$ | VS | $VS_{reg,TvA}$ | DC | $DC_{reg,TvA}$ | HB | $HB_{TvA}$ | Iso | $Iso_{TvA}$ | Beta | $Beta_{TvA}$ | BBQ | $BBQ_{TvA}$ |
|---|---|---|---|---|---|---|---|---|---|---|---|---|---|---|---|---|
| VGG16 | 2.69 | **0.53** | 1.84 | 1.83 | 1.69 | 1.95 | 4.92 | 4.61 | 8.48 | 0.79 | 4.01 | 0.92 | 3.33 | 1.13 | 8.93 | 0.87 |
| Mean improvement VGG | | | -1% | | 16% | | -6% | | **-91%** | | -77% | | -66% | | -90% | |
| ResNet-18 | 2.72 | **0.57** | 1.88 | 1.88 | 1.76 | 2.13 | 3.51 | 3.63 | 9.04 | 0.87 | 3.87 | 0.93 | 3.29 | 1.22 | 9.68 | 0.90 |
| ResNet-34 | 3.63 | **0.62** | 1.78 | 1.81 | 1.85 | 2.01 | 3.51 | 3.13 | 8.61 | 0.71 | 4.08 | 0.84 | 3.81 | 1.07 | 9.17 | 0.87 |
| ResNet-50 | 41.1 | 2.62 | 3.23 | 1.61 | 3.28 | 0.94 | 3.26 | 0.94 | 4.66 | **0.48** | 4.67 | 0.67 | 4.73 | 2.05 | 8.44 | 0.66 |
| ResNet-101 | 13.6 | **0.46** | 3.76 | 2.24 | 4.23 | 1.58 | 4.21 | 1.57 | 5.64 | 0.62 | 3.01 | 0.71 | 4.22 | 1.66 | 6.35 | 0.61 |
| Mean improvement ResNet | | | -22% | | -26% | | -35% | | -90% | | -69% | | -63% | | **-91%** | |
| EffNet-B7 | 12.6 | **0.39** | 3.69 | 2.97 | 3.85 | 1.37 | 3.86 | 1.40 | 4.35 | 0.52 | 2.93 | 0.65 | 5.46 | 1.85 | 6.93 | 0.58 |
| EffNetV2-S | 16.9 | **0.50** | 3.57 | 3.34 | 3.92 | 1.45 | 3.92 | 1.44 | 4.65 | 0.56 | 2.97 | 0.67 | 5.30 | 2.24 | 7.66 | 0.68 |
| EffNetV2-M | 24.9 | 0.72 | 3.73 | 2.69 | 3.84 | 1.16 | 3.84 | 1.16 | 3.97 | **0.55** | | | 4.44 | 1.32 | 6.54 | 0.76 |
| EffNetV2-L | 8.48 | **0.37** | 2.83 | 1.32 | 3.08 | 1.03 | 3.04 | 1.02 | 4.29 | 0.43 | 2.51 | 0.65 | 3.87 | 1.02 | 5.06 | 0.54 |
| Mean improvement EffNet | | | -27% | | -66% | | -66% | | -88% | | -76% | | -67% | | **-90%** | |
| ConvNeXt-T | 16.9 | 0.89 | 3.03 | 1.46 | 3.49 | 1.17 | 3.49 | 1.15 | 5.13 | **0.69** | 2.55 | 0.87 | 3.26 | 1.33 | 7.34 | 0.70 |
| ConvNeXt-S | 17.6 | 0.62 | 3.71 | 2.27 | 4.18 | 1.32 | 4.18 | 1.30 | 4.77 | **0.60** | 3.06 | 0.70 | 4.41 | 1.64 | 7.46 | 0.68 |
| ConvNeXt-B | 18.8 | **0.42** | 3.80 | 2.48 | 4.09 | 1.32 | 4.11 | 1.38 | 4.45 | 0.59 | 3.03 | 0.77 | 4.35 | 1.53 | 7.72 | 0.70 |
| ConvNeXt-L | 12.5 | 0.50 | 4.03 | 2.68 | 4.43 | 1.66 | 4.43 | 1.64 | 4.21 | **0.49** | 3.26 | 0.67 | 4.93 | 1.27 | 7.12 | 0.62 |
| Mean improvement ConvNeXt | | | -40% | | -66% | | -66% | | -87% | | -74% | | -65% | | **-91%** | |
| ViT-B/32 | 6.37 | **0.52** | 3.97 | 2.17 | 4.68 | 1.80 | 4.65 | 1.78 | 7.67 | 0.72 | 3.58 | 0.84 | 4.45 | 1.40 | 9.51 | 0.73 |
| ViT-B/16 | 5.61 | **0.52** | 3.77 | 3.24 | 4.29 | 1.97 | 4.28 | 1.89 | 6.13 | 0.62 | 3.39 | 0.79 | 5.32 | 1.88 | 5.88 | 0.71 |
| ViT-L/32 | 4.27 | 0.73 | 5.01 | 3.89 | 5.37 | 2.54 | 5.37 | 2.51 | 7.71 | **0.64** | 4.43 | 0.76 | 5.75 | 2.04 | 9.31 | 0.79 |
| ViT-L/16 | 5.17 | 0.80 | 5.76 | 4.65 | 5.28 | 2.62 | 5.27 | 2.59 | 7.51 | **0.70** | 4.10 | 0.85 | 6.89 | 2.87 | 6.83 | 0.78 |
| ViT-H/14 | 0.60 | 0.48 | 1.84 | 0.89 | 1.95 | 1.21 | 2.00 | 1.23 | 2.83 | **0.47** | 2.47 | 0.62 | 5.38 | 0.49 | 1.67 | 0.63 |
| Mean improvement ViT | | | -31% | | -51% | | -52% | | **-89%** | | -78% | | -69% | | -85% | |
| Swin-T | 6.82 | 0.51 | 3.08 | 1.82 | 3.44 | 1.39 | 3.43 | 1.34 | 5.51 | **0.50** | 2.94 | 0.72 | 4.14 | 1.15 | 6.72 | 0.67 |
| Swin-S | 3.65 | **0.61** | 3.63 | 2.93 | 4.18 | 1.77 | 4.17 | 1.73 | 4.66 | 0.70 | 3.29 | 0.77 | 5.30 | 1.92 | 7.20 | 0.80 |
| Swin-B | 4.77 | **0.46** | 3.90 | 3.45 | 4.21 | 1.99 | 4.21 | 1.95 | 4.03 | 0.61 | 3.33 | 0.75 | 5.81 | 1.95 | 6.83 | 0.68 |
| SwinV2-T | 8.31 | **0.49** | 3.58 | 2.24 | 3.92 | 1.53 | 3.92 | 1.47 | 5.33 | 0.57 | 3.08 | 0.81 | 4.53 | 1.39 | 7.81 | 0.79 |
| SwinV2-S | 6.07 | **0.46** | 3.77 | 3.32 | 4.25 | 1.73 | 4.24 | 1.72 | 4.42 | 0.53 | 3.16 | 0.74 | 5.28 | 2.00 | 7.18 | 0.67 |
| SwinV2-B | 5.50 | **0.45** | 3.78 | 3.68 | 4.25 | 1.78 | 4.23 | 1.76 | 3.82 | 0.56 | 3.34 | 0.67 | 5.43 | 2.25 | 6.78 | 0.63 |
| Mean improvement Swin | | | -21% | | -58% | | -59% | | -87% | | -77% | | -65% | | **-90%** | |

**Models** Because we study post-processing calibration, we use pre-trained models. For CIFAR, we use the architectures ResNet (He et al., 2016), Wide-ResNet-26-10 (WRN) (Zagoruyko & Komodakis, 2016), and DenseNet-121 (Huang et al., 2017). Weights come from (Mukhoti et al., 2020), following training with the Brier loss as it gives better-calibrated models than those trained with the cross-entropy loss. For ImageNet, we use the architectures VGG (Simonyan & Zisserman, 2015), ResNet, ViT (Dosovitskiy et al., 2020), ConvNeXt (Liu et al., 2022b), EfficientNet (Tan & Le, 2019; 2021), and Swin (Liu et al., 2021; 2022a). We use the models and weights from torchvision.

**Baselines** Our top-versus-all ($_{TvA}$) reformulation and regularization ($_{reg}$) can be applied to different calibration methods. We have tested with scaling methods: Temperature Scaling (TS), Vector Scaling (VS), and Dirichlet Calibration (DC). We also considered binary methods: Histogram Binning (HB), Isotonic Regression (Iso), Beta Calibration (Beta), and Bayesian Binning into Quantiles (BBQ). For Dirichlet calibration, we use the best-performing variant Dir-ODIR, which regularizes off-diagonal and bias coefficients. We also include I-Max binning with shared class-wise strategy

Patel et al. (2020), a state-of-the-art method for ImageNet. This method is a competitive approach to ours because it reformulates the problem of calibrating multiclass classifiers using a shared class-wise strategy instead of our top-versus-all approach. The method includes an optimization process to minimize accuracy degradation, while our approach preserves the accuracy of binary methods. More implementation details can be found in A.3.

**Metrics**  Our main metric is the ECE (2) with 15 equal-width bins, which is the most widely used in the literature. We do not use classwise metrics because we tackle the confidence calibration problem and because they do not scale well when per-class test data is scarce ($\approx 25$ for ImageNet). We discuss this issue in more detail in the appendix A.1.

### 4.1 TOP-VERSUS-ALL

For visual qualitative results, we display a few reliability diagrams (Niculescu-Mizil & Caruana, 2005) in Figure 1. We observe that originally ResNet-50 is highly underconfident and ViT-B/16 slightly underconfident. Applying TS and VS solves the underconfidence and even makes the models slightly overconfident. TvA further improves these methods, and the average confidence and accuracy are much closer. $HB_{TvA}$ is even better, and miscalibration is almost invisible on the diagrams.

Table 1 shows the results of applying the top-versus-all reformulation to several calibration methods. Experiments results are averaged over 5 different random seeds that generate different calibration datasets. In most cases, the TvA reformulation significantly lowers the ECE by dozens of percent. Without TvA, binary methods often perturb the prediction and degrade the classifier's accuracy by more than 1%, making them inapplicable in a practical setting. TvA solves the issue as it only scales the confidence (after the prediction is made) and makes binary methods outperform scaling methods. We also found that Dirichlet calibration is sensitive to hyperparameter tuning, and its performance is usually not much better than vector scaling. This observation is consistent with the results in Kull et al. (2019). ECE computed with equal-mass bins gives similar values as seen in Table 5.

The most competitive approach, I-Max, also has some of the best results. Compared to our approach, it is more complex (our approach does not modify the histogram binning algorithm) and may change the prediction. When the initial calibration error is large (e.g. ResNet-50 on ImageNet in table 1), its calibration tuning is less effective than ours. In addition, Lin et al. (2022) found that I-Max produces unusable probability vectors (they do not sum to 1, and normalizing them degrades the method's performance).

In most cases, TvA also lowers the Brier score, except for Isotonic regression, which has the lowest Brier score overall. The appendix shows full results in Table 6.

We observed that ImageNet networks are mostly underconfident, in accordance with Galil et al. (2023). This observation goes against previous knowledge on overconfidence, which was believed to be linked to network size (Guo et al., 2017). Full results can be seen in Table 3 in the appendix.

To summarize the results for practical use, our experiments show that histogram binning (within the TvA or I-Max setting) is the best calibration method overall and the one we advise to use. However, if the underlying application requires a confidence with continuous values, e.g., to rank the predictions in the case of selective classification, then we advise using a method that also improves the AUROC, shown in A.2, such as temperature scaling or isotonic regression.

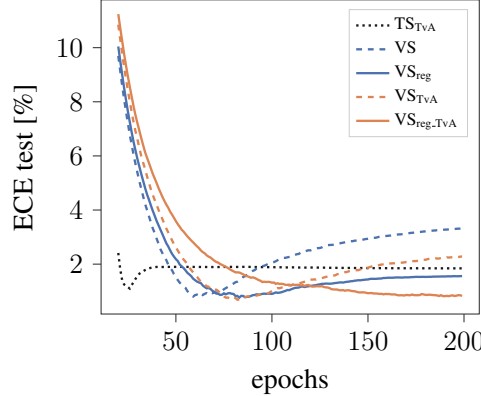

Figure 2: ECE evolution during training with ResNet-50 on ImageNet. Combining regularization and TvA prevents overfitting.

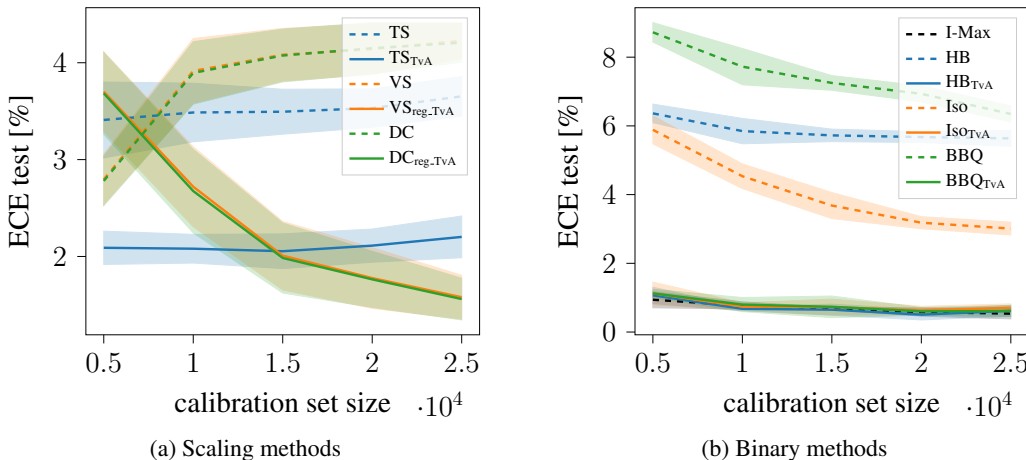

Figure 3: Influence of the calibration set size for ResNet-101 on ImageNet.

## 4.2 Solving overfitting with regularization and TvA

On ImageNet, vector scaling and Dirichlet calibration seem to overfit the calibration set, degrading the calibration on the test set. The lower performance of vector scaling relative to temperature scaling indicates this overfitting. As visualized in Figure 2, combining the binary cross entropy loss used in the TvA reformulation and an additional regularization term prevents overfitting. We fixed the value $\lambda = 0.01$ as it works well across models. We also found that initializing the vector coefficients (or diagonal coefficient for Dirichlet) to $\frac{1}{T}$ with $T$ obtained by temperature sampling (with TvA) helps further improve performance. On the other hand, penalizing values far from $\frac{1}{T}$ instead of 1 degrades the performance.

## 4.3 Influence of the calibration set size

The size of the calibration set influences the performance of the different methods, as seen in Figure 3. Temperature scaling (original and top-versus-all) does not require much data and does not benefit from more data due to its low expressiveness. Vector scaling and Dirichlet calibration do not improve because of the overfitting problem. With regularization and TvA, vector scaling and Dirichlet calibration benefit from more calibration data. With enough data ($\approx 15000$), they outperform temperature scaling. Binary methods (histogram binning, isotonic regression, and Bayesian binning with quantiles; beta calibration was omitted for figure clarity) using the standard one-versus-all approach have bad performance and need a large amount of data to be competitive. With the TvA reformulation, they get great performance with little data.

## 5 Conclusion

Reducing the miscalibration of neural networks is important to improve the trust in their predictions. This can be done after a model is trained with an optimization using calibration data. However, many current calibration methods do not scale to more complex datasets: binary methods under the one-versus-all setting do not have enough per-class calibration data, and scaling methods with many parameters overfit the calibration data. We solve the overfitting issue of vector scaling by adding a regularization term. We demonstrate that reformulating the confidence calibration of multiclass classifiers as a single binary problem significantly improves the performance of baseline calibration techniques. Our TvA reformulation increases the competitiveness of scaling methods and allows binary methods to efficiently use per-class calibration data without altering the model accuracy. In short, it enhances many existing calibration algorithms without modifying them. Extensive experiments on state-of-the-art classification models on ImageNet demonstrate the scalability of our approach. Our TvA reformulation could be a basis for developing future methods that further improve confidence calibration for classifiers with many classes.

ACKNOWLEDGMENTS

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

## A   APPENDIX

A.1 discusses the limits of classwise-ECE and top-label-ECE for a high number of classes.
A.2 shows the impact of different calibration methods on selective classification.
A.3 describes implementation details.
A.4 displays additional results. Table 3 reports the confidences, Table 4 the accuracies, Table 5 the ECE with equal-mass bins, Table 6 the Brier score.

### A.1   LIMITS OF CLASSWISE-ECE AND TOP-LABEL-ECE FOR A HIGH NUMBER OF CLASSES

Let us define the ECE for class $j$:

$$\text{ECE}_j = \sum_{b=1}^{B} \frac{n_b}{N} |\text{acc}(b,j) - \text{conf}(b,j)|$$

The difference compared to 2 is that now $\text{acc}(b,j)$ corresponds to the accuracy of class $j$ in the bin $b$: the proportion of samples predicted as $j$ which actually are $j$. Also, $\text{conf}(b,j)$ now is the average probability given to class $j$ for all samples in the bin.
Then, classwise-ECE (Kull et al., 2019) takes the average for all classes:

$$\text{ECE}_{\text{cw}} = \sum_{j=1}^{L} \text{ECE}_j$$

Classwise-ECE considers the full probabilities vectors: all the class probabilities for each prediction. This metric does not scale to large numbers of classes. Let us see why with an example.
Let us use a test set of 25000 samples, 25 for each of the 1000 ImageNet classes, and a high-accuracy classifier fairly calibrated. For class $j$, there are 25000 predicted probabilities: around 25 of which are close to 1 (the correct predictions, for which confidence is usually high because accuracy is high), and the 24975 remaining are mostly close to 0 (because the predicted class is not class $j$, and the number of classes is high). Now those samples are partitioned into 15 equal mass bins of 1666 samples. For the last bin (with the highest confidences), because of the high imbalance between the 25 positive samples and the rest, the average probability is $\approx 0$, and class occurrence is $\approx 0$. For all the other bins, the average probability is also $\approx 0$, and the class occurrence is $\approx 0$. When the number of classes is high, because the classifier predictions with high confidence do not significantly impact the metric, classwise ECE only measures whereas confidences close to 0 are calibrated. We argue this is not what we are interested in: what matters more is the calibration of the predicted class, which mostly corresponds to high values of confidence. Nixon et al. (2019); Patel et al. (2020) discuss thresholding to filter out small probabilities. However, the value of the threshold has to be tuned, which makes comparisons with other works more difficult.

Top-label ECE (Gupta & Ramdas, 2022) is another interesting metric that does not scale to large numbers of classes either. Top-label-ECE divides data into subsets according to the predicted class, computes the ECEs of these subsets, and averages them. For an ImageNet test set of 25000 samples (25 per class), data is divided into 1000 subsets of $\approx 25$ samples each (the classifier is high-accuracy, most of the time class predicted = true class). The ECE is computed for each subset containing only 25 samples. To compute the ECE, samples are typically partitioned into 15 bins. The number of samples per bin does not allow a correct estimation of the average confidence or accuracy.

### A.2   IMPACT ON SELECTIVE CLASSIFICATION

Selective classification aims to improve a model's prediction performance by trading-off coverage: a reject option allows to discard data that might result in wrong predictions, thus improving the accuracy on the remaining data. A strong standard baseline uses thresholding on the maximum softmax probability outputted by the classifier (Geifman & El-Yaniv, 2017). Improving confidence calibration means uncertainty is better quantified and should result in better selective classification.

Results in Table 1 show the superiority of histogram binning (applied with the right framework) in reducing the calibration error ECE. Unfortunately, it does not translate into improvements in selective classification. AUROC is a standard metric for selective classification (Galil et al., 2023). Table 2 shows that histogram binning actually degrades the AUROC, while the best method is isotonic regression. Our TvA framework does not significantly impact the AUROC.

Table 2: AUROC in % (higher is better). Methods in purple impact the model prediction, potentially degrading accuracy; methods in teal do not. Improvements from the uncalibrated model are colored in blue while degradations are colored in orange

(a) CIFAR-10

| Model | Uncal. | I-Max | TS | TS$_{TvA}$ | VS | VS$_{reg.TvA}$ | DC | DC$_{reg.TvA}$ | HB | HB$_{TvA}$ | Iso | Iso$_{TvA}$ | Beta | Beta$_{TvA}$ | BBQ | BBQ$_{TvA}$ |
|---|---|---|---|---|---|---|---|---|---|---|---|---|---|---|---|---|
| ResNet-50 | 91.12 | 90.05 | 91.04 | 91.01 | 91.67 | 91.67 | 91.66 | 91.65 | 78.92 | 90.16 | 90.79 | 90.72 | 90.74 | 91.12 | 75.19 | 83.32 |
| ResNet-110 | 92.32 | 91.58 | 92.24 | 92.22 | 92.21 | 92.43 | 92.19 | 92.43 | 76.46 | 90.48 | 92.34 | 92.21 | 90.64 | 92.32 | 76.66 | 84.00 |
| WRN | 91.08 | 89.97 | 91.11 | 91.09 | 91.38 | 92.34 | 91.39 | 92.34 | 77.74 | 90.24 | 91.93 | 90.93 | 90.95 | 91.08 | 76.78 | 86.41 |
| DenseNet | 90.03 | 89.29 | 90.08 | 90.04 | 90.24 | 90.83 | 90.26 | 90.83 | 80.03 | 88.71 | 91.20 | 89.28 | 89.76 | 90.03 | 72.70 | 86.90 |

(b) CIFAR-100

| Model | Uncal. | I-Max | TS | TS$_{TvA}$ | VS | VS$_{reg.TvA}$ | DC | DC$_{reg.TvA}$ | HB | HB$_{TvA}$ | Iso | Iso$_{TvA}$ | Beta | Beta$_{TvA}$ | BBQ | BBQ$_{TvA}$ |
|---|---|---|---|---|---|---|---|---|---|---|---|---|---|---|---|---|
| ResNet-50 | 85.55 | 84.93 | 85.48 | 85.36 | 85.69 | 85.99 | 85.71 | 86.01 | 81.60 | 85.40 | 86.90 | 85.46 | 85.47 | 85.55 | 81.19 | 85.15 |
| ResNet-110 | 84.83 | 83.64 | 84.74 | 84.65 | 84.65 | 85.09 | 84.62 | 85.06 | 82.24 | 84.64 | 86.34 | 84.79 | 84.56 | 84.83 | 79.86 | 84.50 |
| WRN | 87.98 | 87.19 | 87.97 | 87.85 | 87.98 | 88.14 | 87.99 | 88.18 | 84.03 | 87.71 | 88.88 | 87.86 | 87.57 | 87.98 | 83.33 | 86.76 |
| DenseNet | 86.61 | 85.54 | 86.56 | 86.42 | 86.85 | 87.36 | 86.86 | 87.38 | 84.12 | 86.43 | 87.57 | 86.56 | 86.53 | 86.61 | 83.06 | 85.91 |

(c) ImageNet

| Model | Uncal. | I-Max | TS | TS$_{TvA}$ | VS | VS$_{reg.TvA}$ | DC | DC$_{reg.TvA}$ | HB | HB$_{TvA}$ | Iso | Iso$_{TvA}$ | Beta | Beta$_{TvA}$ | BBQ | BBQ$_{TvA}$ |
|---|---|---|---|---|---|---|---|---|---|---|---|---|---|---|---|---|
| VGG16 | 86.68 | 86.50 | 86.61 | 86.61 | 86.34 | 86.41 | 85.44 | 85.55 | 83.96 | 86.52 | 86.73 | 86.67 | 85.99 | 86.68 | 83.04 | 86.28 |
| ResNet-18 | 85.73 | 85.40 | 85.64 | 85.65 | 85.65 | 85.88 | 85.16 | 85.28 | 83.98 | 85.60 | 86.11 | 85.69 | 85.31 | 85.73 | 83.19 | 85.43 |
| ResNet-34 | 86.18 | 85.82 | 86.11 | 86.10 | 86.24 | 86.41 | 85.91 | 86.11 | 83.24 | 86.01 | 86.40 | 86.14 | 85.78 | 86.18 | 82.26 | 85.81 |
| ResNet-50 | 80.53 | 80.06 | 85.92 | 85.69 | 85.62 | 85.60 | 85.59 | 85.58 | 83.42 | 80.30 | 86.91 | 80.48 | 83.00 | 80.53 | 85.27 | 80.52 |
| ResNet-101 | 84.18 | 83.62 | 85.96 | 85.71 | 85.39 | 85.55 | 85.37 | 85.55 | 81.92 | 84.03 | 87.09 | 84.18 | 83.58 | 84.18 | 82.48 | 84.13 |
| EffNet-B7 | 84.92 | 84.14 | 86.61 | 86.34 | 85.19 | 85.51 | 85.21 | 85.51 | 80.99 | 84.69 | 87.14 | 84.87 | 83.59 | 84.92 | 81.57 | 84.95 |
| EffNetV2-S | 85.77 | 84.82 | 87.02 | 86.86 | 85.30 | 85.65 | 85.31 | 85.62 | 81.16 | 85.55 | 87.42 | 85.74 | 84.10 | 85.77 | 82.44 | 85.77 |
| EffNetV2-M | 82.36 | 81.59 | 85.26 | 84.92 | 83.65 | 84.17 | 83.65 | 84.20 | 79.78 | 82.11 | 86.51 | 82.32 | 81.42 | 82.36 | 81.24 | 82.24 |
| EffNetV2-L | 84.63 | 83.98 | 86.33 | 86.04 | 85.76 | 85.95 | 85.72 | 85.92 | 80.73 | 84.30 | 86.70 | 84.58 | 83.51 | 84.63 | 81.78 | 84.55 |
| ConvNeXt-T | 82.35 | 81.72 | 85.46 | 85.16 | 85.61 | 85.57 | 85.60 | 85.58 | 81.94 | 82.08 | 86.97 | 82.29 | 82.83 | 82.35 | 82.58 | 82.31 |
| ConvNeXt-S | 82.29 | 81.88 | 85.26 | 84.87 | 84.78 | 85.03 | 84.81 | 85.03 | 80.69 | 82.06 | 86.98 | 82.23 | 82.20 | 82.29 | 81.29 | 82.20 |
| ConvNeXt-B | 82.27 | 81.74 | 85.12 | 84.74 | 84.41 | 84.88 | 84.43 | 84.90 | 80.38 | 82.01 | 87.01 | 82.25 | 82.08 | 82.27 | 81.79 | 82.24 |
| ConvNeXt-L | 82.35 | 81.47 | 84.82 | 84.38 | 84.04 | 84.59 | 84.05 | 84.59 | 79.78 | 82.11 | 86.79 | 82.34 | 81.67 | 82.35 | 80.49 | 82.23 |
| ViT-B/32 | 85.57 | 85.09 | 86.30 | 86.13 | 85.95 | 85.97 | 85.93 | 85.96 | 83.07 | 85.40 | 87.16 | 85.55 | 84.73 | 85.57 | 83.39 | 85.56 |
| ViT-B/16 | 85.52 | 84.81 | 86.32 | 86.12 | 85.36 | 85.56 | 85.36 | 85.53 | 81.39 | 85.32 | 87.19 | 85.48 | 83.78 | 85.52 | 81.56 | 85.34 |
| ViT-L/32 | 85.42 | 84.73 | 85.93 | 85.73 | 85.19 | 85.29 | 85.20 | 85.30 | 81.44 | 85.29 | 87.25 | 85.41 | 83.81 | 85.42 | 81.51 | 85.40 |
| ViT-L/16 | 85.85 | 84.86 | 86.16 | 86.00 | 84.33 | 84.66 | 84.31 | 84.68 | 80.10 | 85.62 | 86.97 | 85.82 | 83.27 | 85.85 | 79.76 | 85.68 |
| ViT-H/14 | 87.28 | 86.44 | 87.53 | 87.34 | 86.74 | 86.73 | 86.79 | 86.77 | 79.99 | 86.94 | 86.66 | 87.21 | 84.62 | 87.28 | 79.33 | 86.55 |
| Swin-T | 85.68 | 85.13 | 86.50 | 86.34 | 85.77 | 85.84 | 85.78 | 85.87 | 81.72 | 85.48 | 87.10 | 85.66 | 84.48 | 85.68 | 82.15 | 85.63 |
| Swin-S | 85.37 | 84.77 | 85.99 | 85.78 | 85.01 | 85.22 | 85.01 | 85.20 | 80.48 | 85.22 | 86.92 | 85.37 | 83.42 | 85.37 | 80.25 | 85.28 |
| Swin-B | 84.11 | 83.39 | 85.26 | 84.91 | 83.90 | 84.19 | 83.92 | 84.19 | 79.41 | 84.12 | 86.55 | 84.17 | 81.95 | 84.11 | 78.94 | 84.18 |
| SwinV2-T | 85.80 | 85.17 | 86.74 | 86.55 | 85.81 | 86.06 | 85.82 | 86.02 | 81.56 | 85.57 | 87.29 | 85.77 | 84.55 | 85.80 | 82.58 | 85.76 |
| SwinV2-S | 85.75 | 85.08 | 86.61 | 86.38 | 85.19 | 85.52 | 85.20 | 85.49 | 80.75 | 85.58 | 87.18 | 85.75 | 83.78 | 85.75 | 80.54 | 85.79 |
| SwinV2-B | 85.15 | 84.15 | 86.07 | 85.82 | 84.42 | 84.68 | 84.40 | 84.69 | 79.75 | 84.95 | 86.99 | 85.14 | 82.42 | 85.15 | 79.57 | 85.02 |

## A.3 IMPLEMENTATION DETAILS

- We used the library netcal (Küppers et al., 2020) for their implementation of binary methods and adapted their reliability diagrams code.
- We took inspiration from the official implementation of temperature scaling: `https://github.com/gpleiss/temperature_scaling`.
- We took inspiration from the official implementation of Dirichlet calibration: `https://github.com/dirichletcal/experiments_dnn`.
- We used the official implementation of I-Max: `https://github.com/boschresearch/imax-calibration`.
- For evaluation, we used codes from `https://github.com/JeremyNixon/uncertainty-metrics-1` and `https://github.com/IdoGalil/benchmarking-uncertainty-estimation-performance`.
- We used PyTorch 2.0.0 and model weights are from torchvision 0.15 `https://github.com/pytorch/vision`.
- We used CIFAR models and weights from Mukhoti et al. (2020) `https://github.com/torrvision/focal_calibration`.

## A.4 ADDITIONAL RESULTS

Table 3: Average confidence in %. Methods in purple impact the model prediction, potentially degrading accuracy; methods in teal do not. Overconfidence (average confidence > accuracy) is shown in violet and underconfidence (average confidence < accuracy) in brown.

### (a) CIFAR-10

| Model | Acc. | Uncal. | I-Max | TS | TS$_{TvA}$ | VS | VS$_{reg,TvA}$ | DC | DC$_{reg,TvA}$ | HB | HB$_{TvA}$ | Iso | Iso$_{TvA}$ | Beta | Beta$_{TvA}$ | BBQ | BBQ$_{TvA}$ |
|---|---|---|---|---|---|---|---|---|---|---|---|---|---|---|---|---|---|
| ResNet-50 | 95.0 | 96.7 | 94.7 | 95.8 | 95.4 | 95.9 | 95.7 | 95.9 | 95.7 | 95.0 | 94.7 | 95.3 | 94.8 | 97.2 | 95.5 | 94.7 | 94.8 |
| ResNet-110 | 94.5 | 97.1 | 94.6 | 95.7 | 95.3 | 95.7 | 95.7 | 95.7 | 95.7 | 95.1 | 94.5 | 95.3 | 94.6 | 97.2 | 94.7 | 95.0 | 94.8 |
| WRN | 95.9 | 96.0 | 95.9 | 96.7 | 96.3 | 96.7 | 96.2 | 96.7 | 96.2 | 96.2 | 95.8 | 96.4 | 95.8 | 97.2 | 96.3 | 95.9 | 95.9 |
| DenseNet | 94.9 | 95.7 | 95.0 | 96.1 | 95.7 | 96.1 | 95.6 | 96.1 | 95.6 | 95.5 | 95.0 | 95.6 | 95.1 | 97.0 | 95.2 | 95.2 | 95.1 |

### (b) CIFAR-100

| Model | Acc. | Uncal. | I-Max | TS | TS$_{TvA}$ | VS | VS$_{reg,TvA}$ | DC | DC$_{reg,TvA}$ | HB | HB$_{TvA}$ | Iso | Iso$_{TvA}$ | Beta | Beta$_{TvA}$ | BBQ | BBQ$_{TvA}$ |
|---|---|---|---|---|---|---|---|---|---|---|---|---|---|---|---|---|---|
| ResNet-50 | 76.6 | 82.7 | 76.6 | 80.7 | 76.6 | 81.1 | 77.2 | 81.0 | 77.2 | 76.4 | 76.5 | 81.2 | 76.5 | 79.9 | 77.4 | 73.8 | 76.5 |
| ResNet-110 | 74.9 | 82.6 | 75.0 | 79.2 | 75.1 | 79.7 | 75.8 | 79.6 | 75.8 | 73.8 | 74.8 | 80.0 | 74.8 | 78.6 | 75.4 | 71.6 | 74.8 |
| WRN | 79.4 | 82.8 | 79.0 | 82.5 | 78.9 | 82.7 | 79.2 | 82.7 | 79.2 | 78.8 | 79.2 | 82.9 | 79.2 | 80.9 | 78.8 | 76.1 | 79.2 |
| DenseNet | 76.2 | 81.1 | 76.1 | 79.8 | 76.1 | 80.2 | 76.6 | 80.2 | 76.6 | 74.9 | 76.1 | 80.2 | 76.1 | 78.9 | 76.0 | 72.6 | 76.1 |

### (c) ImageNet

| Model | Acc. | Uncal. | I-Max | TS | TS$_{TvA}$ | VS | VS$_{reg,TvA}$ | DC | DC$_{reg,TvA}$ | HB | HB$_{TvA}$ | Iso | Iso$_{TvA}$ | Beta | Beta$_{TvA}$ | BBQ | BBQ$_{TvA}$ |
|---|---|---|---|---|---|---|---|---|---|---|---|---|---|---|---|---|---|
| VGG16 | 71.5 | 74.0 | 70.3 | 71.2 | 71.3 | 72.2 | 72.3 | 73.8 | 72.5 | 71.6 | 71.6 | 75.0 | 71.6 | 70.2 | 71.5 | 67.7 | 71.6 |
| ResNet-18 | 69.8 | 72.0 | 69.3 | 69.4 | 69.7 | 70.3 | 70.8 | 71.4 | 70.8 | 69.3 | 69.8 | 73.1 | 69.8 | 68.3 | 69.6 | 65.3 | 69.8 |
| ResNet-34 | 73.2 | 76.8 | 72.9 | 73.4 | 73.1 | 74.4 | 74.3 | 75.3 | 74.2 | 73.7 | 73.3 | 76.8 | 73.3 | 72.1 | 73.1 | 70.3 | 73.3 |
| ResNet-50 | 80.8 | 39.7 | 77.9 | 84.0 | 82.2 | 84.2 | 80.2 | 84.2 | 80.2 | 76.6 | 80.9 | 81.6 | 80.9 | 76.1 | 81.1 | 71.1 | 80.9 |
| ResNet-101 | 81.9 | 68.3 | 81.5 | 85.6 | 83.4 | 86.0 | 83.1 | 86.0 | 83.1 | 81.5 | 82.0 | 84.5 | 82.0 | 79.6 | 81.9 | 78.2 | 82.0 |
| EffNet-B7 | 85.8 | 71.6 | 84.0 | 87.8 | 85.7 | 88.3 | 85.6 | 88.3 | 85.5 | 84.7 | 84.1 | 87.0 | 84.0 | 81.0 | 84.1 | 82.2 | 84.1 |
| EffNetV2-S | 84.2 | 67.3 | 84.0 | 87.8 | 85.6 | 88.2 | 85.3 | 88.2 | 85.2 | 84.3 | 84.2 | 86.8 | 84.2 | 81.4 | 84.4 | 81.2 | 84.2 |
| EffNetV2-M | 84.3 | 60.2 | 84.7 | 88.8 | 86.5 | 89.1 | 85.9 | 89.1 | 85.8 | 85.2 | 85.2 | 87.7 | 85.2 | 81.9 | 85.2 | 81.8 | 85.2 |
| EffNetV2-L | 85.1 | 77.3 | 85.4 | 88.6 | 86.9 | 88.9 | 86.7 | 88.9 | 86.6 | 86.6 | 85.6 | 88.1 | 85.6 | 83.5 | 85.6 | 84.2 | 85.6 |
| ConvNeXt-T | 82.5 | 65.6 | 81.7 | 85.5 | 83.9 | 85.9 | 83.3 | 85.9 | 83.2 | 82.1 | 82.5 | 84.7 | 82.5 | 79.8 | 82.6 | 78.9 | 82.5 |
| ConvNeXt-S | 83.6 | 66.0 | 83.2 | 87.3 | 85.2 | 87.8 | 84.7 | 87.8 | 84.7 | 83.7 | 83.7 | 86.3 | 83.6 | 80.4 | 83.9 | 80.7 | 83.6 |
| ConvNeXt-B | 84.0 | 65.3 | 83.7 | 87.8 | 85.6 | 88.2 | 85.0 | 88.2 | 85.0 | 84.0 | 84.0 | 86.7 | 84.0 | 81.6 | 84.1 | 81.2 | 84.1 |
| ConvNeXt-L | 84.4 | 71.9 | 84.3 | 88.4 | 86.2 | 88.8 | 85.8 | 88.8 | 85.8 | 84.9 | 84.5 | 87.4 | 84.5 | 81.9 | 84.7 | 82.6 | 84.5 |
| ViT-B/32 | 75.9 | 69.6 | 75.6 | 79.9 | 77.2 | 80.4 | 77.4 | 80.4 | 77.3 | 75.1 | 75.9 | 78.9 | 75.9 | 74.7 | 75.6 | 71.0 | 75.9 |
| ViT-B/16 | 81.0 | 75.5 | 81.0 | 84.8 | 82.6 | 85.3 | 82.8 | 85.3 | 82.7 | 81.5 | 81.0 | 84.0 | 81.0 | 79.5 | 81.1 | 78.8 | 81.0 |
| ViT-L/32 | 77.0 | 74.2 | 77.2 | 81.6 | 78.8 | 82.2 | 79.0 | 82.2 | 79.0 | 77.1 | 76.9 | 80.8 | 76.9 | 76.9 | 76.9 | 73.9 | 76.9 |
| ViT-L/16 | 79.6 | 78.8 | 80.0 | 84.4 | 81.6 | 85.1 | 82.1 | 85.1 | 82.1 | 80.5 | 79.7 | 83.6 | 79.7 | 78.2 | 79.9 | 78.3 | 79.7 |
| ViT-H/14 | 88.6 | 89.0 | 88.5 | 90.4 | 89.3 | 90.5 | 89.5 | 90.5 | 89.5 | 89.8 | 88.6 | 90.8 | 88.6 | 83.7 | 88.5 | 88.7 | 88.6 |
| Swin-T | 81.5 | 74.7 | 81.1 | 84.5 | 82.5 | 85.0 | 82.7 | 85.0 | 82.6 | 81.8 | 81.5 | 84.0 | 81.5 | 79.5 | 81.4 | 78.8 | 81.5 |
| Swin-S | 83.2 | 79.9 | 83.1 | 86.8 | 84.6 | 87.3 | 84.7 | 87.3 | 84.7 | 83.8 | 83.2 | 86.1 | 83.1 | 81.2 | 83.0 | 81.7 | 83.1 |
| Swin-B | 83.6 | 79.7 | 83.7 | 87.5 | 85.4 | 88.0 | 85.6 | 88.0 | 85.5 | 84.5 | 83.5 | 86.7 | 83.5 | 81.2 | 83.6 | 82.8 | 83.5 |
| SwinV2-T | 82.0 | 73.7 | 81.9 | 85.6 | 83.3 | 86.0 | 83.4 | 86.0 | 83.4 | 82.3 | 82.2 | 84.7 | 82.2 | 79.6 | 82.2 | 79.4 | 82.2 |
| SwinV2-S | 83.7 | 77.7 | 83.7 | 87.5 | 85.2 | 88.0 | 85.4 | 88.0 | 85.4 | 84.6 | 83.7 | 86.7 | 83.7 | 82.3 | 83.7 | 82.3 | 83.7 |
| SwinV2-B | 84.1 | 78.9 | 84.1 | 87.9 | 85.7 | 88.4 | 85.8 | 88.4 | 85.8 | 85.0 | 84.1 | 87.1 | 84.2 | 82.2 | 84.0 | 83.1 | 84.1 |

Table 4: Accuracy in % (higher is better). Methods in purple impact the model prediction, potentially degrading accuracy; methods in teal do not. Because classifiers can be well calibrated when not accurate (by having low accuracy and low confidence), it is important to monitor the accuracy. It is even better when the methods preserve the accuracy by design.

(a) CIFAR-10

| Model | Uncal. | I-Max | TS | $TS_{TvA}$ | VS | $VS_{reg.TvA}$ | DC | $DC_{reg.TvA}$ | HB | $HB_{TvA}$ | Iso | $Iso_{TvA}$ | Beta | $Beta_{TvA}$ | BBQ | $BBQ_{TvA}$ |
|---|---|---|---|---|---|---|---|---|---|---|---|---|---|---|---|---|
| ResNet-50 | 95.00 | 94.99 | 95.00 | 95.00 | 94.94 | 94.96 | 94.95 | 94.97 | 94.49 | 95.00 | 94.82 | 95.00 | 94.96 | 95.00 | 94.65 | 95.00 |
| ResNet-110 | 94.52 | 94.50 | 94.52 | 94.52 | 94.44 | 94.42 | 94.45 | 94.41 | 94.13 | 94.52 | 94.40 | 94.52 | 94.45 | 94.52 | 94.18 | 94.52 |
| WRN | 95.92 | 95.87 | 95.92 | 95.92 | 95.86 | 95.84 | 95.86 | 95.84 | 95.76 | 95.92 | 95.80 | 95.92 | 95.84 | 95.92 | 95.78 | 95.92 |
| DenseNet | 94.89 | 94.92 | 94.89 | 94.89 | 94.98 | 94.99 | 94.98 | 94.99 | 94.59 | 94.89 | 94.82 | 94.89 | 94.92 | 94.89 | 94.71 | 94.89 |

(b) CIFAR-100

| Model | Uncal. | I-Max | TS | $TS_{TvA}$ | VS | $VS_{reg.TvA}$ | DC | $DC_{reg.TvA}$ | HB | $HB_{TvA}$ | Iso | $Iso_{TvA}$ | Beta | $Beta_{TvA}$ | BBQ | $BBQ_{TvA}$ |
|---|---|---|---|---|---|---|---|---|---|---|---|---|---|---|---|---|
| ResNet-50 | 76.61 | 76.59 | 76.61 | 76.61 | 76.37 | 76.36 | 76.38 | 76.35 | 74.10 | 76.61 | 76.06 | 76.61 | 76.52 | 76.61 | 75.92 | 76.61 |
| ResNet-110 | 74.90 | 74.86 | 74.90 | 74.90 | 74.60 | 74.73 | 74.64 | 74.75 | 72.15 | 74.90 | 74.36 | 74.90 | 74.75 | 74.90 | 74.17 | 74.90 |
| WRN | 79.41 | 79.37 | 79.41 | 79.41 | 79.12 | 79.06 | 79.11 | 79.03 | 77.00 | 79.41 | 78.80 | 79.41 | 79.26 | 79.41 | 78.54 | 79.41 |
| DenseNet | 76.25 | 76.42 | 76.25 | 76.25 | 76.18 | 76.14 | 76.18 | 76.13 | 74.03 | 76.25 | 75.82 | 76.25 | 76.26 | 76.25 | 75.60 | 76.25 |

(c) ImageNet

| Model | Uncal. | I-Max | TS | $TS_{TvA}$ | VS | $VS_{reg.TvA}$ | DC | $DC_{reg.TvA}$ | HB | $HB_{TvA}$ | Iso | $Iso_{TvA}$ | Beta | $Beta_{TvA}$ | BBQ | $BBQ_{TvA}$ |
|---|---|---|---|---|---|---|---|---|---|---|---|---|---|---|---|---|
| VGG16 | 71.54 | 70.34 | 71.54 | 71.54 | 71.53 | 70.89 | 68.96 | 68.05 | 68.15 | 71.54 | 71.00 | 71.54 | 71.65 | 71.54 | 70.30 | 71.54 |
| ResNet-18 | 69.77 | 69.37 | 69.77 | 69.77 | 69.79 | 69.22 | 68.05 | 67.51 | 66.01 | 69.77 | 69.24 | 69.77 | 69.99 | 69.77 | 68.34 | 69.77 |
| ResNet-34 | 73.23 | 72.73 | 73.23 | 73.23 | 73.20 | 72.68 | 71.91 | 71.32 | 69.87 | 73.23 | 72.81 | 73.23 | 73.30 | 73.23 | 72.21 | 73.23 |
| ResNet-50 | 80.85 | 80.26 | 80.85 | 80.85 | 80.92 | 80.79 | 80.94 | 80.79 | 78.20 | 80.85 | 80.47 | 80.85 | 80.79 | 80.85 | 78.13 | 80.85 |
| ResNet-101 | 81.86 | 81.52 | 81.86 | 81.86 | 81.77 | 81.65 | 81.78 | 81.64 | 79.32 | 81.86 | 81.44 | 81.86 | 81.88 | 81.86 | 80.82 | 81.86 |
| EffNet-B7 | 84.16 | 84.04 | 84.16 | 84.16 | 84.44 | 84.30 | 84.43 | 84.30 | 82.23 | 84.16 | 84.09 | 84.16 | 84.36 | 84.16 | 83.71 | 84.16 |
| EffNetV2-S | 84.27 | 84.02 | 84.27 | 84.27 | 84.33 | 84.24 | 84.32 | 84.24 | 82.30 | 84.27 | 83.88 | 84.27 | 84.34 | 84.27 | 83.52 | 84.27 |
| EffNetV2-M | 85.06 | 84.91 | 85.06 | 85.06 | 85.28 | 85.19 | 85.28 | 85.18 | 83.39 | 85.06 | 84.87 | 85.06 | 85.17 | 85.06 | 84.19 | 85.06 |
| EffNetV2-L | 85.80 | 85.64 | 85.80 | 85.80 | 85.89 | 85.83 | 85.92 | 85.84 | 83.99 | 85.80 | 85.58 | 85.80 | 86.00 | 85.80 | 85.23 | 85.80 |
| ConvNeXt-T | 82.50 | 82.19 | 82.50 | 82.50 | 82.43 | 82.29 | 82.44 | 82.28 | 79.94 | 82.50 | 82.10 | 82.50 | 82.51 | 82.50 | 81.51 | 82.50 |
| ConvNeXt-S | 83.65 | 83.38 | 83.65 | 83.65 | 83.64 | 83.54 | 83.63 | 83.54 | 81.33 | 83.65 | 83.28 | 83.65 | 83.67 | 83.65 | 82.89 | 83.65 |
| ConvNeXt-B | 84.04 | 83.78 | 84.04 | 84.04 | 84.09 | 83.98 | 84.08 | 83.97 | 81.88 | 84.04 | 83.68 | 84.04 | 84.13 | 84.04 | 83.22 | 84.04 |
| ConvNeXt-L | 84.38 | 84.25 | 84.38 | 84.38 | 84.41 | 84.31 | 84.41 | 84.31 | 82.44 | 84.38 | 84.12 | 84.38 | 84.49 | 84.38 | 83.98 | 84.38 |
| ViT-B/32 | 75.95 | 75.69 | 75.95 | 75.95 | 75.81 | 75.66 | 75.82 | 75.66 | 72.66 | 75.95 | 75.36 | 75.95 | 75.98 | 75.95 | 74.59 | 75.95 |
| ViT-B/16 | 81.04 | 80.88 | 81.04 | 81.04 | 81.00 | 80.87 | 81.00 | 80.90 | 78.56 | 81.04 | 80.63 | 81.04 | 81.08 | 81.04 | 80.38 | 81.04 |
| ViT-L/32 | 76.96 | 76.83 | 76.96 | 76.96 | 76.80 | 76.73 | 76.79 | 76.71 | 74.20 | 76.96 | 76.37 | 76.96 | 76.98 | 76.96 | 76.04 | 76.96 |
| ViT-L/16 | 79.64 | 79.55 | 79.64 | 79.64 | 79.81 | 79.67 | 79.82 | 79.66 | 77.44 | 79.64 | 79.47 | 79.64 | 79.76 | 79.64 | 79.20 | 79.64 |
| ViT-H/14 | 88.62 | 88.54 | 88.62 | 88.62 | 88.62 | 88.49 | 88.58 | 88.46 | 87.00 | 88.62 | 88.34 | 88.62 | 88.68 | 88.62 | 88.33 | 88.62 |
| Swin-T | 81.49 | 81.27 | 81.49 | 81.49 | 81.56 | 81.42 | 81.56 | 81.39 | 78.96 | 81.49 | 81.07 | 81.49 | 81.58 | 81.49 | 80.77 | 81.49 |
| Swin-S | 83.21 | 83.04 | 83.21 | 83.21 | 83.12 | 83.01 | 83.13 | 83.03 | 81.02 | 83.21 | 82.79 | 83.21 | 83.26 | 83.21 | 82.74 | 83.21 |
| Swin-B | 83.60 | 83.51 | 83.60 | 83.60 | 83.77 | 83.60 | 83.77 | 83.60 | 81.69 | 83.60 | 83.39 | 83.60 | 83.72 | 83.60 | 83.40 | 83.60 |
| SwinV2-T | 82.02 | 81.81 | 82.02 | 82.02 | 82.13 | 81.97 | 82.12 | 82.00 | 79.69 | 82.02 | 81.66 | 82.02 | 82.18 | 82.02 | 81.27 | 82.02 |
| SwinV2-S | 83.74 | 83.65 | 83.74 | 83.74 | 83.80 | 83.71 | 83.80 | 83.72 | 82.03 | 83.74 | 83.56 | 83.74 | 83.82 | 83.74 | 83.34 | 83.74 |
| SwinV2-B | 84.10 | 84.02 | 84.10 | 84.10 | 84.14 | 84.07 | 84.16 | 84.05 | 82.27 | 84.10 | 83.81 | 84.10 | 84.18 | 84.10 | 83.79 | 84.10 |

Table 5: ECE with 15 equal mass bins in % (lower is better). Methods in purple impact the model prediction, potentially degrading accuracy; methods in teal do not.

(a) CIFAR-10

| Model | Uncal. | I-Max | TS | TS$_{TvA}$ | VS | VS$_{reg.TvA}$ | DC | DC$_{reg.TvA}$ | HB | HB$_{TvA}$ | Iso | Iso$_{TvA}$ | Beta | Beta$_{TvA}$ | BBQ | BBQ$_{TvA}$ |
|---|---|---|---|---|---|---|---|---|---|---|---|---|---|---|---|---|
| ResNet-50 | 1.74 | 1.05 | 1.25 | 1.12 | 1.28 | 1.21 | 1.27 | 1.20 | 2.21 | 1.27 | 1.30 | **1.01** | 2.27 | 1.42 | 1.96 | 1.09 |
| ResNet-110 | 2.61 | 0.59 | 1.74 | 1.65 | 1.70 | 1.77 | 1.68 | 1.77 | 2.69 | **0.39** | 1.02 | 0.62 | 2.79 | 1.74 | 2.61 | 1.12 |
| WRN | 1.69 | **0.10** | 1.63 | 1.64 | 1.23 | 1.34 | 1.23 | 1.35 | 1.86 | 0.43 | 0.79 | 0.58 | 1.62 | 1.82 | 1.54 | 0.23 |
| DenseNet | 1.98 | 0.45 | 1.94 | 1.97 | 1.48 | 1.65 | 1.48 | 1.65 | 2.70 | **0.35** | 0.83 | 0.71 | 2.12 | 2.24 | 2.39 | 0.81 |

(b) CIFAR-100

| Model | Uncal. | I-Max | TS | TS$_{TvA}$ | VS | VS$_{reg.TvA}$ | DC | DC$_{reg.TvA}$ | HB | HB$_{TvA}$ | Iso | Iso$_{TvA}$ | Beta | Beta$_{TvA}$ | BBQ | BBQ$_{TvA}$ |
|---|---|---|---|---|---|---|---|---|---|---|---|---|---|---|---|---|
| ResNet-50 | 6.58 | **0.78** | 4.92 | 3.40 | 5.08 | 2.17 | 5.06 | 2.17 | 10.88 | 1.70 | 5.33 | 1.57 | 5.32 | 3.68 | 9.84 | 1.99 |
| ResNet-110 | 7.73 | **1.17** | 5.12 | 3.94 | 5.15 | 3.00 | 5.13 | 2.98 | 11.06 | 1.76 | 6.08 | 1.74 | 5.48 | 4.96 | 9.51 | 1.25 |
| WRN | 4.20 | **0.74** | 4.03 | 2.75 | 4.22 | 1.98 | 4.21 | 1.95 | 10.38 | 1.10 | 4.41 | 0.89 | 4.26 | 2.90 | 9.76 | 1.60 |
| DenseNet | 5.08 | **0.98** | 4.03 | 2.24 | 4.31 | 2.05 | 4.29 | 2.03 | 10.39 | 1.03 | 4.42 | 1.35 | 4.65 | 2.67 | 10.71 | 1.42 |

(c) ImageNet

| Model | Uncal. | I-Max | TS | TS$_{TvA}$ | VS | VS$_{reg.TvA}$ | DC | DC$_{reg.TvA}$ | HB | HB$_{TvA}$ | Iso | Iso$_{TvA}$ | Beta | Beta$_{TvA}$ | BBQ | BBQ$_{TvA}$ |
|---|---|---|---|---|---|---|---|---|---|---|---|---|---|---|---|---|
| VGG16 | 2.62 | **0.47** | 1.84 | 1.81 | 1.60 | 1.89 | 4.84 | 4.55 | 9.71 | 0.80 | 4.00 | 0.92 | 3.31 | 1.05 | 9.86 | 0.80 |
| ResNet-18 | 2.59 | **0.54** | 1.86 | 1.83 | 1.69 | 2.08 | 3.43 | 3.59 | 9.63 | 0.89 | 3.85 | 0.75 | 3.21 | 1.03 | 9.84 | 0.66 |
| ResNet-34 | 3.61 | **0.58** | 1.75 | 1.75 | 1.79 | 1.97 | 3.43 | 3.05 | 9.68 | 0.76 | 4.04 | 0.72 | 3.71 | 1.04 | 9.17 | 0.68 |
| ResNet-50 | 41.15 | 2.63 | 3.17 | 1.74 | 3.25 | 1.11 | 3.23 | 1.12 | 4.63 | **0.64** | 1.29 | 0.76 | 4.89 | 2.04 | 8.24 | 0.98 |
| ResNet-101 | 13.55 | **0.46** | 3.73 | 2.35 | 4.21 | 1.58 | 4.19 | 1.55 | 6.62 | 0.73 | 3.01 | 0.64 | 4.08 | 1.94 | 7.84 | 1.00 |
| EffNet-B7 | 12.60 | **0.46** | 3.82 | 2.94 | 3.83 | 1.60 | 3.84 | 1.56 | 6.13 | 0.71 | 2.93 | 0.56 | 5.32 | 1.96 | 6.91 | 0.71 |
| EffNetV2-S | 16.92 | **0.39** | 4.02 | 3.32 | 3.91 | 1.67 | 3.91 | 1.67 | 6.20 | 0.72 | 2.96 | 0.65 | 5.13 | 2.15 | 7.58 | 0.92 |
| EffNetV2-M | 24.88 | **0.72** | 3.74 | 2.66 | 3.84 | 1.35 | 3.83 | 1.35 | 5.20 | 0.77 | 2.88 | 0.73 | 4.36 | 1.46 | 6.68 | 1.01 |
| EffNetV2-L | 8.48 | **0.43** | 2.81 | 1.47 | 3.05 | 0.93 | 3.03 | 0.93 | 5.24 | 0.70 | 2.51 | 0.62 | 3.72 | 1.21 | 6.03 | 0.78 |
| ConvNeXt-T | 16.95 | 0.95 | 3.03 | 1.62 | 3.48 | 1.19 | 3.47 | 1.21 | 6.11 | 0.85 | 2.55 | **0.82** | 3.25 | 1.49 | 7.64 | 0.99 |
| ConvNeXt-S | 17.60 | **0.58** | 3.76 | 2.55 | 4.17 | 1.44 | 4.17 | 1.43 | 6.06 | 0.76 | 3.06 | 0.71 | 4.28 | 1.90 | 7.36 | 0.79 |
| ConvNeXt-B | 18.77 | **0.46** | 3.78 | 2.67 | 4.08 | 1.48 | 4.09 | 1.49 | 5.86 | 0.78 | 3.03 | 0.73 | 4.25 | 1.97 | 7.48 | 1.04 |
| ConvNeXt-L | 12.51 | **0.43** | 4.03 | 2.89 | 4.42 | 1.84 | 4.42 | 1.83 | 6.02 | 0.64 | 3.26 | 0.63 | 4.96 | 1.52 | 7.05 | 0.78 |
| ViT-B/32 | 6.37 | **0.50** | 4.06 | 2.46 | 4.64 | 1.89 | 4.62 | 1.85 | 8.83 | 0.76 | 3.58 | 0.71 | 4.33 | 1.48 | 9.24 | 0.74 |
| ViT-B/16 | 5.56 | **0.50** | 4.17 | 3.18 | 4.27 | 2.12 | 4.26 | 2.03 | 7.66 | 0.76 | 3.38 | 0.70 | 5.25 | 1.99 | 7.68 | 0.93 |
| ViT-L/32 | 4.13 | 0.73 | 5.31 | 4.20 | 5.37 | 2.67 | 5.37 | 2.66 | 8.95 | 0.82 | 4.42 | **0.72** | 5.81 | 2.33 | 9.18 | 1.04 |
| ViT-L/16 | 5.17 | **0.64** | 5.92 | 5.19 | 5.28 | 2.75 | 5.26 | 2.74 | 8.79 | 0.87 | 4.10 | 0.76 | 6.97 | 3.09 | 7.82 | 1.29 |
| ViT-H/14 | 0.61 | **0.42** | 1.76 | 0.84 | 1.88 | 1.10 | 1.95 | 1.10 | 4.58 | 0.58 | 2.45 | 0.60 | 5.21 | 0.49 | 3.95 | 0.53 |
| Swin-T | 6.82 | **0.53** | 3.09 | 1.81 | 3.42 | 1.33 | 3.42 | 1.28 | 6.97 | 0.63 | 2.94 | 0.71 | 4.07 | 1.30 | 7.52 | 0.90 |
| Swin-S | 3.57 | **0.58** | 3.92 | 2.98 | 4.18 | 1.84 | 4.17 | 1.79 | 7.06 | 0.76 | 3.29 | 0.59 | 5.24 | 2.05 | 6.78 | 0.83 |
| Swin-B | 4.65 | **0.35** | 4.37 | 3.72 | 4.21 | 2.09 | 4.20 | 2.04 | 6.74 | 0.80 | 3.33 | 0.63 | 5.84 | 2.42 | 6.50 | 0.78 |
| SwinV2-T | 8.31 | **0.49** | 3.59 | 2.19 | 3.91 | 1.60 | 3.92 | 1.55 | 6.80 | 0.66 | 3.07 | 0.71 | 4.48 | 1.48 | 7.97 | 0.67 |
| SwinV2-S | 6.06 | **0.44** | 4.17 | 3.32 | 4.24 | 1.90 | 4.24 | 1.85 | 6.63 | 0.72 | 3.15 | 0.56 | 5.33 | 2.09 | 7.04 | 0.79 |
| SwinV2-B | 5.27 | **0.37** | 4.40 | 3.67 | 4.24 | 1.93 | 4.22 | 1.92 | 6.55 | 0.68 | 3.33 | 0.57 | 5.56 | 2.34 | 6.56 | 0.75 |

Table 6: Brier score of the predicted class in $10^{-2}$ (lower is better). Methods in purple impact the model prediction, potentially degrading accuracy; methods in teal do not.

(a) CIFAR-10

| Model | Uncal. | I-Max | TS | TS$_{TvA}$ | VS | VS$_{reg.TvA}$ | DC | DC$_{reg.TvA}$ | HB | HB$_{TvA}$ | Iso | Iso$_{TvA}$ | Beta | Beta$_{TvA}$ | BBQ | BBQ$_{TvA}$ |
|---|---|---|---|---|---|---|---|---|---|---|---|---|---|---|---|---|
| ResNet-50 | 3.79 | 3.76 | 3.74 | **3.72** | 3.74 | 3.73 | 3.74 | 3.73 | 4.02 | 3.78 | 3.77 | 3.75 | 3.91 | 3.76 | 4.01 | 3.84 |
| ResNet-110 | 4.02 | 3.76 | 3.84 | 3.81 | 3.81 | 3.81 | 3.81 | 3.81 | 4.31 | 3.76 | **3.72** | 3.74 | 4.07 | 3.84 | 4.25 | 4.02 |
| WRN | 3.02 | **2.95** | 3.06 | 3.03 | 3.09 | 3.03 | 3.09 | 3.03 | 3.25 | 3.03 | **2.95** | 2.97 | 3.11 | 3.06 | 3.22 | 3.03 |
| DenseNet | 3.75 | **3.64** | 3.78 | 3.76 | 3.72 | 3.70 | 3.72 | 3.69 | 4.08 | 3.68 | 3.65 | **3.64** | 3.85 | 3.77 | 4.13 | 3.67 |

(b) CIFAR-100

| Model | Uncal. | I-Max | TS | TS$_{TvA}$ | VS | VS$_{reg.TvA}$ | DC | DC$_{reg.TvA}$ | HB | HB$_{TvA}$ | Iso | Iso$_{TvA}$ | Beta | Beta$_{TvA}$ | BBQ | BBQ$_{TvA}$ |
|---|---|---|---|---|---|---|---|---|---|---|---|---|---|---|---|---|
| ResNet-50 | 12.86 | 12.13 | 12.55 | 12.26 | 12.56 | 12.18 | 12.55 | 12.17 | 13.73 | 12.15 | 12.21 | **12.12** | 12.57 | 12.29 | 14.30 | 12.24 |
| ResNet-110 | 13.89 | 12.84 | 13.30 | 12.98 | 13.35 | 12.90 | 13.36 | 12.91 | 14.18 | 12.82 | 12.98 | **12.77** | 13.33 | 13.11 | 15.09 | 12.84 |
| WRN | 11.00 | 10.73 | 10.97 | 10.80 | 10.97 | 10.76 | 10.96 | 10.74 | 12.27 | 10.68 | 10.78 | **10.65** | 11.03 | 10.74 | 12.69 | 10.78 |
| DenseNet | 12.29 | 12.02 | 12.12 | 11.93 | 12.18 | 11.80 | 12.17 | **11.79** | 13.23 | 11.87 | 11.89 | 11.84 | 12.20 | 11.91 | 13.99 | 11.91 |

(c) ImageNet

| Model | Uncal. | I-Max | TS | TS$_{TvA}$ | VS | VS$_{reg.TvA}$ | DC | DC$_{reg.TvA}$ | HB | HB$_{TvA}$ | Iso | Iso$_{TvA}$ | Beta | Beta$_{TvA}$ | BBQ | BBQ$_{TvA}$ |
|---|---|---|---|---|---|---|---|---|---|---|---|---|---|---|---|---|
| VGG16 | 13.18 | 13.15 | 13.18 | 13.18 | 13.29 | 13.39 | 14.46 | 14.52 | 14.78 | 13.15 | 13.32 | 13.11 | 13.36 | **13.11** | 15.09 | 13.15 |
| ResNet-18 | 13.93 | **13.84** | 13.94 | 13.93 | 13.92 | 13.92 | 14.57 | 14.61 | 15.20 | 13.89 | 13.89 | 13.87 | 14.02 | 13.86 | 15.56 | 13.90 |
| ResNet-34 | 13.15 | 12.99 | 13.04 | 13.05 | 12.98 | 13.04 | 13.53 | 13.53 | 14.58 | 13.03 | 13.13 | 12.99 | 13.15 | **12.98** | 14.92 | 13.01 |
| ResNet-50 | 29.79 | 12.33 | 10.95 | 10.88 | 10.98 | 10.91 | 10.98 | 10.92 | 11.46 | 12.08 | **10.71** | 12.02 | 11.69 | 12.06 | 11.90 | 12.08 |
| ResNet-101 | 12.65 | 10.77 | 10.66 | 10.51 | 10.70 | 10.51 | 10.70 | 10.51 | 11.48 | 10.75 | **10.35** | 10.71 | 10.84 | 10.75 | 11.72 | 10.74 |
| EffNet-B7 | 11.28 | 9.68 | 9.71 | 9.55 | 9.72 | 9.51 | 9.73 | 9.51 | 10.50 | 9.66 | **9.41** | 9.60 | 9.94 | 9.67 | 10.71 | 9.62 |
| EffNetV2-S | 12.39 | 9.50 | 9.66 | 9.48 | 9.71 | 9.50 | 9.71 | 9.50 | 10.43 | 9.48 | **9.37** | 9.43 | 9.79 | 9.51 | 10.64 | 9.46 |
| EffNetV2-M | 16.05 | 9.82 | 9.58 | 9.43 | 9.54 | 9.33 | 9.54 | 9.32 | 10.13 | 9.78 | **9.18** | 9.72 | 9.80 | 9.75 | 10.40 | 9.75 |
| EffNetV2-L | 9.80 | 9.07 | 8.90 | 8.81 | 8.93 | 8.83 | 8.94 | 8.84 | 9.85 | 9.05 | **8.81** | 9.00 | 9.11 | 9.01 | 9.96 | 9.01 |
| ConvNeXt-T | 14.02 | 10.97 | 10.39 | 10.33 | 10.40 | 10.33 | 10.40 | 10.32 | 11.28 | 10.94 | **10.13** | 10.87 | 10.73 | 10.88 | 11.61 | 10.90 |
| ConvNeXt-S | 13.62 | 10.35 | 10.15 | 10.01 | 10.16 | 9.95 | 10.16 | 9.95 | 10.88 | 10.37 | **9.78** | 10.32 | 10.40 | 10.36 | 11.27 | 10.35 |
| ConvNeXt-B | 13.86 | 10.20 | 10.04 | 9.89 | 10.01 | 9.79 | 10.01 | 9.79 | 10.65 | 10.21 | **9.60** | 10.14 | 10.22 | 10.18 | 11.11 | 10.18 |
| ConvNeXt-L | 11.58 | 9.97 | 9.92 | 9.76 | 9.99 | 9.70 | 9.99 | 9.69 | 10.54 | 9.96 | **9.46** | 9.88 | 10.15 | 9.91 | 10.96 | 9.90 |
| ViT-B/32 | 12.68 | 12.26 | 12.34 | 12.17 | 12.53 | 12.33 | 12.53 | 12.33 | 13.57 | 12.27 | **12.11** | 12.24 | 12.60 | 12.25 | 13.72 | 12.25 |
| ViT-B/16 | 11.02 | 10.71 | 10.88 | 10.72 | 11.02 | 10.83 | 11.01 | 10.83 | 12.08 | 10.70 | **10.59** | 10.66 | 11.14 | 10.69 | 12.09 | 10.67 |
| ViT-L/32 | 12.15 | 12.02 | 12.35 | 12.11 | 12.49 | 12.21 | 12.49 | 12.21 | 13.56 | 11.98 | **11.92** | 11.93 | 12.64 | 11.98 | 13.87 | 11.96 |
| ViT-L/16 | 11.39 | 11.18 | 11.73 | 11.46 | 11.88 | 11.57 | 11.88 | 11.56 | 12.71 | 11.17 | 11.13 | **11.11** | 12.06 | 11.23 | 13.21 | 11.15 |
| ViT-H/14 | 7.46 | 7.51 | 7.49 | 7.46 | 7.57 | 7.58 | 7.58 | 7.59 | 8.53 | 7.54 | 7.58 | 7.47 | 7.94 | **7.46** | 8.60 | 7.48 |
| Swin-T | 11.09 | 10.65 | 10.63 | 10.53 | 10.72 | 10.64 | 10.71 | 10.63 | 11.89 | 10.65 | **10.51** | 10.61 | 10.82 | 10.62 | 11.89 | 10.63 |
| Swin-S | 10.14 | 10.03 | 10.23 | 10.06 | 10.32 | 10.12 | 10.31 | 10.12 | 11.22 | 10.03 | **9.95** | 9.98 | 10.47 | 10.03 | 11.48 | 10.00 |
| Swin-B | 10.18 | 10.00 | 10.21 | 10.05 | 10.23 | 10.10 | 10.23 | 10.10 | 11.07 | 9.99 | **9.82** | 9.91 | 10.52 | 9.98 | 11.46 | 9.93 |
| SwinV2-T | 11.06 | 10.37 | 10.45 | 10.29 | 10.54 | 10.37 | 10.54 | 10.38 | 11.58 | 10.40 | **10.25** | 10.34 | 10.65 | 10.36 | 11.69 | 10.36 |
| SwinV2-S | 10.05 | 9.66 | 9.91 | 9.72 | 10.01 | 9.77 | 10.00 | 9.78 | 10.78 | 9.66 | **9.57** | 9.62 | 10.11 | 9.68 | 11.13 | 9.64 |
| SwinV2-B | 10.00 | 9.71 | 9.96 | 9.79 | 10.04 | 9.83 | 10.04 | 9.82 | 10.86 | 9.70 | **9.59** | 9.64 | 10.22 | 9.72 | 11.21 | 9.66 |

Table 7: Computing time of the calibration; in seconds. The first column denotes the data preprocessing time, which includes computing the model logits for all calibration examples.

(a) ImageNet

| Model | Preproc. | I-Max | TS | TS$_{TvA}$ | VS | VS$_{reg.TvA}$ | DC | DC$_{reg.TvA}$ | HB | HB$_{TvA}$ | Iso | Iso$_{TvA}$ | Beta | Beta$_{TvA}$ | BBQ | BBQ$_{TvA}$ |
|---|---|---|---|---|---|---|---|---|---|---|---|---|---|---|---|---|
| ResNet-50 | 141 | 543 | 215 | 218 | 214 | 217 | 226 | 226 | 129 | 1 | 66 | 1 | 873 | 22 | 1156 | 2 |
| ViT-B/16 | 151 | 524 | 225 | 226 | 217 | 222 | 232 | 235 | 127 | 1 | 61 | 1 | 917 | 23 | 1169 | 2 |

