# OpenReview forum: "Efficient calibration as a binary top-versus-all problem for classifiers with many classes"
_ICLR.cc/2024/Conference — Submitted to ICLR 2024_

### Official Review · Reviewer_75D8 · 2023-10-20

**Soundness:** 3 good
**Presentation:** 3 good
**Contribution:** 2 fair
**Rating:** 5
**Confidence:** 4

**Summary:**

This paper propose tot improve the current calibration methods by convert it into a binary case under the one-versus-all setting and demonstrate that reformulating the confidence calibration of multiclass classifiers as a single binary problem significantly improves the performance of baseline calibration techniques.

**Strengths:**

It study the shortcome of many post hoc calibration problem and provide a better loss to improve post hoc calibration method.

**Weaknesses:**

See questions.

**Questions:**

1. This work provide a good and easy to improve most post hoc method, however it seems too simple. I would say it is more like a part of a post hoc method paper although the author give comprehensive experiments.
2. It would be better to include more metrics other than ECE.
3. I would suggest the author to include the TvA into training time calibration to see if it works.

---

> ### Author Response · Authors · 2023-11-16
> **Response to reviewer 75D8**
>
> Thank you for your feedback. Here are our answers to your questions.
>
> > 1. This work provide a good and easy to improve most post hoc method, however it seems too simple. I would say it is more like a part of a post hoc method paper although the author give comprehensive experiments.
>
> In our experiments, we demonstrate the competitiveness of the method. Combined with its simplicity, it becomes a valuable tool for practical use cases, where a compromise on development cost must be made so the most complicated methods are less easily deployable.
>
>
> > 2. It would be better to include more metrics other than ECE.
>
> We already included results for other metrics in the Appendix. Namely, AUROC (Table 2) for selective classification, ECE with 15 equal mass bins, and Brier score. Also, in the first part of the Appendix (A.1), we also discussed why classwise-ECE and top-label-ECE are unsuitable for multiclass classification with many classes.
>
> > 3. I would suggest the author to include the TvA into training time calibration to see if it works.
>
> Adapting our approach to this training time calibration setting still seems an interesting idea to pursue. However, the paper focuses on post-hoc calibration, where the goal is to calibrate an already trained model. The main advantages are that it improves the use of off-the-shelf models, and it decouples the model training (optimizing for accuracy) and calibration. These advantages significantly lower the development cost to obtain a well-performing and well-calibrated model. That is why we chose to focus on post-hoc calibration. As you mentioned, another family of approaches directly optimizes calibration during model training, potentially leading to better calibration, but is more complex to develop for practical use.

---

> > ### Comment · Reviewer_75D8 · 2023-11-23
> >
> > I have read the authors rebuttal, and still found the paper lack of novelty. I will keep my score.

---

### Official Review · Reviewer_TXd7 · 2023-10-30

**Soundness:** 3 good
**Presentation:** 3 good
**Contribution:** 2 fair
**Rating:** 6
**Confidence:** 4

**Summary:**

The paper addresses the calibration of multi-class classifiers trained to discriminate many classes. The proposal consists in using a binary top-versus-all approach: the calibration problem is transformed into providing a confidence estimate regarding whether the prediction made by the classifier is correct. The authors first provide a succinct state-of-the-art on calibration approaches, then present the classifier calibration problem, and present their contribution. Experimental results are provided, before the paper briefly concludes.

**Strengths:**

The paper is overall written in a clear and understandable manner, and is pleasant to read.

The results displayed in the Experiments section are good and show that the proposal is interesting.

**Weaknesses:**

The contributions of the paper seem somehow rather restricted: the proposal consists in recasting the calibration problem into a binary problem (i.e., adjusting the level of confidence in the prediction issued by the classifier); there is no theoretical study. The proposal is not really formalized.

The state-of-the-art does not include a number of works on classifier calibration, which may have been interesting to include in the discussions and in the experiments (see, e.g., Venn predictors).

Some parts in the paper are redundant—for instance, Sections 2 (related work) and 3 (problem setting) are tightly connected and may have been merged into a single one. Section 4 also mentions some related work which could have been presented and discussed previously. The notations are sometimes inconsistent (e.g., the authors use small x's and y's as well as capital ones interchangeably; as well, they indistinctly use "one-versus-all", "one-vs-the-rest", etc.)

**Questions:**

In Section 2, page 3, you mention "more advanced methods": can you be more specific ? As well, when referring to Gupta and Ramdas (2022) which first defines the top-label calibrator, their work should be presented with more details (here or in the "Problem setting" section) as it is highly connected to the proposal.

In Section 3.2: "This discretizes the probability." This sentence is a bit clumsy; can you clarify ?

In Section 3.3, you may also mention that the ECE is not a proper scoring rule. This also questions its use as a metric for assessing calibration performance. Could you provide any insight regarding this ?

Section 4 should be improved. In its present state, it is hard to see what is exactly the proposal. In particular, I think that the proposed approach should be clearly and formally stated (and not only via Algorithm 1), notably by explicitly formulating the criterion used to replace Equation (1)—this would clarify the difference with the former top-vs-all proposal by Gupta and Ramdas (2022).

In Section 4.1, could you elaborate on "minimizing the cross-entropy loss increases the probability of
the correct class (thus only indirectly decreasing the confidence), but minimizing the binary cross-entropy
loss directly decreases the confidence" ?

In its current state, Section 4.2 is short, which is regrettable since it addresses the more important part in the paper—the properties of the proposal. The argument that "the [proposed] reformulation [of the top-vs-all approach uses] the full calibration dataset" could be discussed: then, the positive and negative classes are imbalanced (and heavily imbalanced in the case of numerous classes), which may degrade performances. Can you discuss this porential issue ? As well, the sentence "the classifier's prediction and accuracy are unaffected" is unclear; in addition, if the classifier's predictions are left unchanged compared to the one-vs-all case, it also means that your approach cannot improve the classifier's accuracy by righting erroneous decisions: can you elaborate on that ?

---

> ### Author Response · Authors · 2023-11-16
> **Response 1/2 to reviewer TXd7**
>
> Thank you for your feedback. Please see below our response.
>
> **Weaknesses:**
>
> > The contributions of the paper seem somehow rather restricted: the proposal consists in recasting the calibration problem into a binary problem (i.e., adjusting the level of confidence in the prediction issued by the classifier); there is no theoretical study. The proposal is not really formalized.
>
> The lack of theory is indeed a weakness of our work. However, we want to highlight that while our approach is conceptually simple, our comprehensive experiments demonstrate excellent results, especially on the challenging ImageNet dataset.
> Our work, conceptual and empiric, suggests that the original formulation of the confidence calibration problem might be flawed. We believe it might inspire subsequent studies more focused on the theory to understand this.
>
> > The state-of-the-art does not include a number of works on classifier calibration, which may have been interesting to include in the discussions and in the experiments (see, e.g., Venn predictors).
>
> Thank you for suggesting the work about Venn predictors. We mentionned it in the related work section of the updated paper.
>
> > Some parts in the paper are redundant—for instance, Sections 2 (related work) and 3 (problem setting) are tightly connected and may have been merged into a single one. Section 4 also mentions some related work which could have been presented and discussed previously.
>
> Indeed, the structure can be improved. This problem was also reported by reviewer fZjD. We merged the contents of sections 2, 3, and 4 into two sections in the updated paper.
>
> > The notations are sometimes inconsistent [...]
>
> Thank you for noting these typos; we fixed them.
>
> **Questions:**
>
> > In Section 2, page 3, you mention "more advanced methods": can you be more specific ? As well, when referring to Gupta and Ramdas (2022) which first defines the top-label calibrator, their work should be presented with more details (here or in the "Problem setting" section) as it is highly connected to the proposal.
>
> Indeed, the word "advanced" is not clear; we meant complex. In the updated paper, we reformulated it as "developed ensemble temperature scaling", which is more specific. We also added details for Gupta and Ramdas (2022) in the related work.
>
> > In Section 3.2: "This discretizes the probability." This sentence is a bit clumsy; can you clarify?
>
> Histogram binning takes a continuous probability value (p) and outputs a discrete value (defined by the bin to which the continuous values belong). For instance, if we have a bin with boundaries [0.8, 0.9) and associated value 0.95, $\forall p \in [0.8, 0.9)$, the calibrated probability is $p_c$ = 0.95.
> We reformulated the sentence as "The probability becomes discrete: it can only take $B$ values", B being the number of bins (which we defined above).
>
> > In Section 3.3, you may also mention that the ECE is not a proper scoring rule. This also questions its use as a metric for assessing calibration performance. Could you provide any insight regarding this ?
>
> You raise the hard question of the evaluation of calibration. ECE is not a proper scoring rule, but to the best of our knowledge, it is the most widely used metric to evaluate (confidence) calibration and compare methods. It comes with a graphical representation (reliability diagrams). We already stated ECE has flaws (e.g., estimation issues due to the binning scheme). A classifier can have a very low ECE, suggesting a good calibration, if it is always wrong with very low confidences. This kind of issue does not happen with our method because we do not directly optimize ECE as a loss function, and we use performant pre-trained classifiers whose prediction is not changed by the calibration (except for vector scaling). Also, in the Appendix, we include results for the Brier score, a proper scoring rule. Thanks to your feedback, we added a mention to the issue you refer to.
>
> > Section 4 should be improved. In its present state, it is hard to see what is exactly the proposal. In particular, I think that the proposed approach should be clearly and formally stated (and not only via Algorithm 1), notably by explicitly formulating the criterion used to replace Equation (1)—this would clarify the difference with the former top-vs-all proposal by Gupta and Ramdas (2022).
>
> We improved the paper to clarify the proposal. Equation (1) defines confidence calibration (often just called calibration), the problem we tackle. It is the same problem as in Guo et al. (2017); we do not replace it. Gupta and Ramdas (2022) introduce another problem: top-label calibration. This is similar to confidence calibration but with additional conditioning on the top-label (predicted class). This notion is interesting and well justified in their paper but does not work when the number of classes is high. The top-label ECE metric cannot properly be estimated in that case. This is what we explained in Appendix A.1.

---

> ### Author Response · Authors · 2023-11-16
> **Response 2/2 to Reviewer TXd7**
>
> > In Section 4.1, could you elaborate on "minimizing the cross-entropy loss increases the probability of the correct class (thus only indirectly decreasing the confidence), but minimizing the binary cross-entropy loss directly decreases the confidence" ?
>
> Let us try to clarify that point. We use different notations from the paper for self-sufficiency and better clarity of the explanation.
>
> The cross-entropy loss is $-\sum_{c=1}^Ly_{c}\cdot\log(p_{c}(x))$ with $L$ the number of classes, $y_c$ a binary indicator (1 if $c$ is the correct class, else 0), $p_c$ the classifier probability for class $c$ and data sample $x$. If the correct class is class $k$, the loss reduces to $-\log(p_k(x))$. Minimizing this loss results in increasing the probability of the true class $p_k(x)$.
>
> There are two cases. Let us denote the confidence as $s(x) = \max_{c} p_c(x)$.
> 1. The prediction is correct, which means $s(x) = p_k(x)$. Minimizing the loss increases $p_k(x)$, which is the confidence.
> 2. The prediction is incorrect, which means $s(x) > p_k(x)$. Minimizing the loss increases $p_k(x)$ again, but does not directly change the confidence (because $s(x) \neq p_k(x)$). Instead, the confidence (which was attributed to a wrong class) is reduced because of the softmax layer: increasing one probability decreases other ones *indirectly*.
>
> In our TvA setting, the binary cross-entropy loss is $- \big(y \cdot \log s(x) + (1-y) \cdot \log (1 - s(x))\big)$ where $y$ is a binary indicator of the prediction correctness (1 when the prediction is correct, else 0), and s(x) the confidence.
> Again, there are two cases:
> 1. The prediction is correct, the loss then reduces to $- \log s(x)$. Minimizing the loss directly increases the confidence (same as for the cross-entropy loss).
> 2. The prediction is incorrect, the loss then reduces to $\log (1 - s(x))$. Minimizing the loss *directly* decreases the confidence. This is a key difference compared to using the cross-entropy loss.
>
> We hope that clarifies the point you raised.
>
> > In its current state, Section 4.2 is short, which is regrettable since it addresses the more important part in the paper—the properties of the proposal. The argument that "the [proposed] reformulation [of the top-vs-all approach uses] the full calibration dataset" could be discussed: then, the positive and negative classes are imbalanced (and heavily imbalanced in the case of numerous classes), which may degrade performances. Can you discuss this porential issue ?
>
> In the standard one-versus-all setting, the calibration problem for L classes is decomposed L binary problems. For each of these problems, the positive examples correspond to the given class $l$, and the negative examples correspond to all the L-1 other classes. The imbalance ratio increases with the number of classes (imbalance ratio 1 vs. 999 for Imagenet).
>
> However, in our top-versus-all setting, remember that the calibration problem becomes a single binary problem, which can be stated as "Is the prediction correct". For this binary problem, the number of positive (resp. negative) examples is the number of examples well-classified (resp. misclassified). The imbalance ratio then only depends on model accuracy. For instance, for 75% accuracy, the ratio is 1 negative for 3 positive examples (imbalance ratio 1 vs. 3). This is much lower than the standard one-versus-all setting and probably one of the main justifications for our approach.
> Indeed, refinements could be made to reduce the impact of this imbalance, but we do not believe it would lead to significant improvements. The imbalance might become a bigger issue for models with almost-perfect accuracies, which is not the case for many complex computer vision tasks, and also, do almost-perfect models need calibration?
>
>
> > As well, the sentence "the classifier's prediction and accuracy are unaffected" is unclear; in addition, if the classifier's predictions are left unchanged compared to the one-vs-all case, it also means that your approach cannot improve the classifier's accuracy by righting erroneous decisions: can you elaborate on that ?
>
> Because post-processing calibration uses a calibration dataset separate from the training dataset, additional information can be gained. As you mention, calibration methods that can change the classifier's prediction might take advantage of this to improve both calibration and accuracy. However, in practice, accuracy is usually degraded, as shown in Table 4 (Beta calibration without TvA is the exception; other ones, such as I-Max or DC mainly reduce accuracy). We also argue that decoupling accuracy improvements (during training time) and calibration (during post-processing calibration) is better. Optimizing the two simultaneously might lead to compromises and require more development and experiments.

---

> > ### Comment · Reviewer_TXd7 · 2023-11-22
> > **Follow-up**
> >
> > Thank you for your answers to my comments and questions. The additional information provided makes the proposal more understandable.
> >
> > On the positive side, the results are nice and show that top-vs-all seemingly improves confidence calibration; the proposal is simple and versatile. On the negative side, the submission lacks a theoretical analysis: this is my major concern.
> >
> > Therefore, I'd be willing to improve my rating, albeit by a small margin.

---

### Official Review · Reviewer_fZjD · 2023-10-31

**Soundness:** 3 good
**Presentation:** 2 fair
**Contribution:** 3 good
**Rating:** 6
**Confidence:** 3

**Summary:**

Paper proposes confidence calibration for multi-class classification as a single binary classification problem using top-vs-all approach. This gives ability to calibrate large number of classes with scarce per-class data, and the usage of binary cross-entropy loss with regularisation term. Benchmark image datasets are utilised to evaluate the proposed approach, showing stability in classification accuracy and calibration improvements against existing methods.

**Strengths:**

Paper proposes sound yet simple idea which improves the existing calibration approaches. It includes good set of experiments and evaluations to show the usefulness of proposed techniques. As model-agnostic approach, it would be possible to apply the algorithm to different existing neural network models and post-processing calibration techniques. This is an interesting idea that could bring some new knowledge to the field, especially from the practical view of uncertainty calibration.

**Weaknesses:**

Background and literature review could be in a more compact form. For now, it is repeated in many sections making the follow of the presentation a bit hard: Otherwise it is clearly written and structured. Paper lacks some of the analysis and discussion of the proposed approach and results in a broader sense. Also, it has limited discussion of the results in relation to practical utilisation of approaches, i.e., which of the proposed combination of algorithms should be selected in different scenarios from practitioners' perspectives. From empirical point of view, it would strengthen the paper, if additional dataset from other than image domain would be considered.

**Questions:**

- References lacks some details, please add all the relevant information to cited work (also for ArXiv pre-prints)
- What would be your conclusions or "rule of thumb" of selecting particular algorithm (i.e., calibration method with TvA) from the practitioners' point of view for certain applications or classification problem?

---

> ### Author Response · Authors · 2023-11-16
> **Response to reviewer fZjD**
>
> Thank you for your feedback. Below is our response to the weaknesses and questions you raised.
>
> **Weaknesses:**
>
> > Background and literature review could be in a more compact form. For now, it is repeated in many sections making the follow of the presentation a bit hard: Otherwise it is clearly written and structured.
>
> Thank you for reporting this issue, which was also reported by reviewer TXd7. We clarified the related work by restructuring the contents of sections 2, 3, and 4 in the updated paper by merging them into two sections.
>
>
> > Paper lacks some of the analysis and discussion of the proposed approach and results in a broader sense.
>
> We updated the paper to include more analysis and discussion of the approach and results.
>
> > Also, it has limited discussion of the results in relation to practical utilisation of approaches, i.e., which of the proposed combination of algorithms should be selected in different scenarios from practitioners' perspectives.
>
> If the goal is the best calibration, we advise using histogram binning (with I-Max or our TvA setting) as it is the best method overall. But if the application requires a finer confidence (continuous values), histogram binning is not recommended as confidences have discrete values (one per bin). In that case, we advise using temperature scaling or isotonic regression as they ensure a good ranking of the predictions, as shown by the AUROC metric in Table 2. We added this discussion at the end of Section 4.1.
>
> > From empirical point of view, it would strengthen the paper, if additional dataset from other than image domain would be considered.
>
> This is a good remark but many closely related work only considers image datasets, such as Gupta and Ramdas (2022), Patel et al. (2022),  Kull et al. (2019) (for experiments on deep neural networks), or Gupta et al. (2021). To compare to these approaches, we also used image datasets. Much of the related work is limited to results on simpler datasets like CIFAR; our comprehensive results on ImageNet are an improvement. Contrary to these works, our experiments also compare many different models. This gives additional insights regarding model architecture's impact on calibration. Overall, vision transformers are better-calibrated out-of-the-box than convolutional neural networks.
>
>
> **Questions:**
>
> > References lacks some details, please add all the relevant information to cited work (also for ArXiv pre-prints)
>
> We added the details for the references in the updated paper.
>
> > What would be your conclusions or "rule of thumb" of selecting particular algorithm (i.e., calibration method with TvA) from the practitioners' point of view for certain applications or classification problem?
>
> Please see our third response in the weaknesses section.

---

> > ### Comment · Reviewer_fZjD · 2023-11-22
> > **Response to rebuttal**
> >
> > Thanks for the clarification. As very empirical-oriented study, it would have been nice to see experiments also with some other classification problems, but I understand that due to space limit, previous benchmarking comparison, and large number of classes (ImageNet), paper concentrates on these particular datasets.
> >
> > In the experiments, proposed TvA approach seems to correct confidence calibration of many multi-class and binary post-processing techniques, showing practical usefulness in the case of large number of classes (which could been seen the main results of the work), although theoretical analyses, why it works, is very limited. I-Max class-wise approach is overall the best or sometimes comparable with TvA (looking at ECE metrics), but I-Max could affect the pre-trained model accuracy. Simplicity and efficiency can been seen as main selling points of proposed approach, so it would be also useful to compare and analyse the computational cost between I-Max and TvA in practice, in relation to performance and robustness of these calibration techniques.

---

> > > ### Author Response · Authors · 2023-11-22
> > > **Response**
> > >
> > > We appreciate you taking the time to read and respond to our response. I-Max, as you note, is the closest competitor to our approach, but it has two limitations: it can affect accuracy, as you noted, and it generates quantized confidence levels, a limitation that also applies to histogram binning. In some applications, a continuous confidence is needed.
> > >
> > > As for the computing time, we added a table at the end of the appendix to compare methods. To summarize, TvA does not accelerate scaling methods (because the procedure is the same, with a different loss) but significantly accelerates binary methods (because we only solve one problem, not one per class). I-Max takes more than 500 seconds, while its closest competitor, histogram binning with TvA, takes less than 1 second.
> > >
> > > If you believe our updated paper and responses clarified some points, we would highly appreciate that you consider raising your rating to increase the chances of the paper's acceptance.

---

> > > > ### Comment · Reviewer_fZjD · 2023-11-23
> > > > **Response**
> > > >
> > > > Thanks for the response and additional experiments with computing times. This clarifies the practical side of the proposal a bit more. Due to limited theoretical support of proposed approach, I'll keep my original score.

---

### Official Review · Reviewer_JkUP · 2023-10-31

**Soundness:** 3 good
**Presentation:** 2 fair
**Contribution:** 2 fair
**Rating:** 3
**Confidence:** 3

**Summary:**

The authors provide a conceptually simple and practical technique for the post calibration of trained models that could scale to the case of
many classes: they reduce the problem to one binary calibration problem (and not many calibration sub problems, such as often is the case in prior work). The paper contains a good discussion of prior work, and presents many empirical experiments and comparisons on vision data sets with up to 1000s of classes.

**Strengths:**

Calibration, or obtaining good or reliable probabilities, is an important task in many areas of machine learning.  Classification into many classes is challenging and occurs often in practice. The authors present a simple problem formulation and reduction that can be practical: that of calibrating the probability assigned to the highest scoring class (the 'confidence'). The paper is fairly clear, and many experiments
and in particular comparisons with other techniques are presented. The authors motivate their approach well (in particular, efficiency
considerations).

**Weaknesses:**

A major issue is weak novelty or contribution.  Another important issue (but somewhat secondary) is that the paper clarity is
somewhat poor too. I'll give a  summary below and then expand on these in the 'Questions' section.

Main issue with contribution: one would think in any practical application of calibration, one would be interested in good
probabilities assigned to top candidates, not just the very top (to make good decisions based on the classifications), but the authors
only focus on the very top in their development of the technique and evaluations (if I am not mistaken, and to keep the solution and the
evaluation simple..). Using the other scores should improve the calibration too.  I believe this severely limits the current contribution, and more research and development of the approach is required to make the paper a technically strong contribution.

**Questions:**

[roughly in order of importance]

With many candidate classes given an instance (eg 100s to 1000s), it
is understandable that one may not want to assign good probabilities
(waste time/resources on) on all the candidates, and focusing on the
top is well motivated (the issue of sparsity of data, for training or
calibrating per class, is understandable as well). However, it is also
not advisable to throw out all the information (all the scores
assigned to the classes), except the top (or the winning) class. For
instance, the spread (closeness) of the scores can be very
informative. Furthermore, in any plausible application of calibration
in this setting, for instance in subsequent decision theoretic
actions, plausibly one wants to know the probabilities assigned to the
other, top few, classes as well.


- not clear how binary methods (such as HB or Iso) are used alone, without TvA for
 calibration.. (eg in Table 1) TvA is used on the top score.. but use of the binary
 methods to this multiclass setting is not clear to me in the experiments.. I don't think the authors explained this.. and then
 the authors explain that the binary methods perturb the decision of the original classifier, etc.

- What is a reference for "I-Max binning".. I believe it is first mentioned on page 7 (from my searching the paper..), and it scores
 very well.. (Table 1, on Imagenet dataset/models) Also: why include it if the probabilities can sum to more than 1.0 with this method ?
 (for some evaluation scores such as log loss, perhaps for ECE too, this could lead to cheating by a method...)

- the authors use 'confidence' (eg on page 4 when they say 'beyond
just considering confidence'), but they define it in passing on page 3
as 'the confidence is the top class probability' (top probability as
opposed to the probability assigned to other, non-top,
classes)... promote it or highlight this technical definition better
(because confidence is a generic term, but here in this paper, at
least from this point on, it has a more technical meaning... at least
after page 3!). For example, the use of 'confidence' in the abstract
(used 3 times) reflects the more generic meaning ...

other clarity comments:

- Intro is vague, for instance, in ".. to predict the true
probability of a good decision, i.e., their accuracy."  What is a
"good decision"? (is it committing to one class or label, for a given
test instance, and the label turning out to be correct? 'accuracy'
often has a technical term in machine learning, which is one minus
zero-one error, or the proportion of test instances correctly
classified.. so if the proportion is 80%, do we want the model to
also always assign a fixed 80% to its classifications? or a
probability that is more fine-grained than that (not fixed at
80%.. which can simply be obtained from cross-validation!) ).. I am
guessing the latter .. perhaps quick/short examples would clarify the statements.  Also the
distinction between 'uncertainty quantification' (at the beginning of
intro) and providing good probabilities or calibration is not clear
either (the techniques are mentioned with citations, but more
explanations would be useful).

-  could drop 'in our work' in "We are interested in our work in.."

- drop 'process' in 'a complementary post-processing
 calibration process'..

- In Related Work section, not sure what 'less complex than the other
 ones' means in the long sentence: 'the problem of confidence calibration, less complex
 than the other ones' (in what ways were the aforementioned citations
 more complex?)

- page 5: "We notice " to "We note " (the former implies you have
 observed something, in your work/experiments, etc. while the latter
 means you want the reader to note or observe something, and that's
 what you mean)

- change "one-vs-the-rest approach" to  "one-vs-rest approach"?

- the semantics of probability P() in equation 1 of 3.1 is not clear
 (in the sense of how it is computed, ie in what way or on what
 probability space, or how is it empirically estimated.. ) ... although
 the example you give afterwards helps. Perhaps insert "(when computed
 on unseen or test instances)" in "the probability of being correct
 when the confidence is ..", so it becomes ".. the probability of
 being correct (when computed on unseen or test instances) when the
 confidence is .."


- Top-versus-Rest (instead of Top-vs-All) ? (I understand one-vs-all
 is commonly used instead of one-vs-rest, and this follows a similar
 pattern)


- 3.3, page 4: the presentation/description of ECE should be
    improved, perhaps by a quick example ..

-  replace 'size' (in 'equal mass or equal size') with 'width'
 perhaps? as 'size' is ambiguous: it could mean bin extent or width,
 or number of points or instances (whose score fall) in the bin (what
 is meant by bin mass, I believe, is number of instances in the bin)..

---

> ### Author Response · Authors · 2023-11-16
> **Response 1/2 to reviewer JkUP**
>
> Thank you for your feedback. thank you for giving us such detailed improvement recommendations. We used them to improve the paper's clarity.
>
> **Weaknesses:**
>
> > A major issue is weak novelty or contribution.
>
> Our idea is simple, efficient, and model-agnostic. This is not just another method for confidence calibration with incremental improvements, we propose a reformulation of the confidence calibration problem. This reformulation significantly improves the performance of calibration baselines (without modifying them) and allows them to scale to complex datasets with many classes. Also, it could be a basis for developing future methods.
> Our comprehensive experiments demonstrate state-of-the-art performance on ImageNet, a complex dataset rarely used for calibration experiments due to its high number of classes.
> On the practical side, our idea only requires a few lines of code to adapt calibration baselines to improve their performance.
>
> > Another important issue (but somewhat secondary) is that the paper clarity is somewhat poor too.
>
> Thank you for providing detailed clarification comments. We updated the paper to improve its clarity by taking into account your remarks, as well as those from the other reviewers.
>
> > Main issue with contribution: one would think in any practical application of calibration, one would be interested in good probabilities assigned to top candidates, not just the very top (to make good decisions based on the classifications), but the authors only focus on the very top in their development of the technique and evaluations (if I am not mistaken, and to keep the solution and the evaluation simple..). Using the other scores should improve the calibration too. I believe this severely limits the current contribution, and more research and development of the approach is required to make the paper a technically strong contribution.
>
> There are different applications. Some might indeed require calibrated probabilities of all the class probabilities (or at least the top candidates). But many applications only use confidence values. For instance, as we mentioned in the article, this is the case for the domains of selective classification (Geifman & El-Yaniv, 2017), out-of-distribution detection (Hendrycks & Gimpel, 2017), or active learning (Li & Sethi, 2006). This recent article ([link](https://www.nature.com/articles/s41586-023-06615-2)) about tumor classification during surgery is a concrete example of a confidence calibration application. Predictions that do not reach the required confidence are rejected (this is selective classification), and the confidence was calibrated using temperature scaling.
>
> For the evaluation, we did not focus on specific metrics for simplicity. We aimed for a comprehensive evaluation. To the best of our knowledge, the ECE is the most widely used metric to assess calibration error and compare methods. What it measures is the confidence calibration error. We also include other metrics in the Appendix (AUROC, ECE with equal-mass bins, and Brier score). Classwise-ECE has been developed to go beyond just considering confidence. Still, this metric and top-label-ECE cannot properly be estimated when the number of classes is high, as we explained in more detail in Appendix A.1.
>
> Using the other scores is what other calibration baselines do (e.g., temperature scaling); this is the standard way of approaching the problem. Our experiments show that our method is better in reducing the calibration error (ECE) and improving the reliability diagrams. We wrote a more detailed explanation as the answer to your first question.

---

> > ### Comment · Reviewer_JkUP · 2023-11-23
> > **I have read all reviews and author responses**
> >
> > I acknowledge and appreciate  the authors' responses, and the work is promising and I like its simplicity.  However, I still believe the work requires quite more to strengthen its technical contributions, in addition to much work needed to polish or improve the quality of  the presentation.
> >
> > My main issue  remains that with many classes, eg on Imagenet, on a given (test) instance, quite often several classes get a reasonable score or probability (eg above 0.1) (the case of a highly isolated top scoring label is not that common..), and the decision theoretic motivation or consideration is to improve calibration of *all  such* (not all classes, but all the few classes near the top..  ).  Perhaps what is lacking is a real application for this multiclass calibration problem...   The authors provided a link to a paper stressing importance of calibration for a diagnosis  problem, "Ultra-fast deep-learned CNS tumour classification...", with a threshold for the classifier staying silent.  However, several classes (labels or courses of action) may make it above that threshold (in the multiclass setting), and  the problem of calibrating all such remains.
> >
> > [ The authors later below state that calibrating all the say 1000 classes can lead to calibration quality degradation for the top, and that is    indeed plausible (in addition to inefficiency issues).   But this does not imply that one should ignore the  classes that score   near the very top. ]

---

> ### Author Response · Authors · 2023-11-16
> **Response 2/2 to reviewer JkUP**
>
> **Questions:**
>
> > With many candidate classes given an instance (eg 100s to 1000s), it is understandable that one may not want to assign good probabilities (waste time/resources on) on all the candidates, and focusing on the top is well motivated (the issue of sparsity of data, for training or calibrating per class, is understandable as well). However, it is also not advisable to throw out all the information (all the scores assigned to the classes), except the top (or the winning) class. For instance, the spread (closeness) of the scores can be very informative. Furthermore, in any plausible application of calibration in this setting, for instance in subsequent decision theoretic actions, plausibly one wants to know the probabilities assigned to the other, top few, classes as well.
>
> We agree that throwing out information except the top class is counterintuitive. And yet, it works. We believe that aiming to calibrate all the probability vector might reduce the importance of calibrating the top class (as all classes are considered equally in the optimization), and thus increase the calibration error for the top class (while in average reducing the calibration error for all classes). Surprisingly, standard confidence calibration methods aim to improve calibration of the confidence only (as measured by the ECE), but optimize the full probability vector, even the components with the lowest values. In our method, we explicitly optimize only for the top class, the one we care about, which may explain the better results.
>
> You are right that some applications of calibration may require calibrated probabilities for the top few or even all classes. However, we argue that confidence calibration is useful for many practical cases, as explained above.
>
> To our knowledge, most studies are about confidence calibration of full vector calibration. It would indeed be interesting to calibrate the top e.g. 5 classes, as a bridge between the two extreme settings. This could be done with our method by reformulating the multiclass problem into 5 top-versus-all problems. We would then derive 5 calibrators, one for each of the top 5 probabilities.
>
> > not clear how binary methods (such as HB or Iso) are used alone, without TvA for calibration.. (eg in Table 1) TvA is used on the top score.. but use of the binary methods to this multiclass setting is not clear to me in the experiments.. I don't think the authors explained this.. and then the authors explain that the binary methods perturb the decision of the original classifier, etc.
>
> Binary methods can be used in the multiclass setting with the one-versus-all approach. We mentioned it in the introduction and explained it in the related work, paragraph Multiclass to Binary. We slightly reformulated this part to make it clearer:
> "The multiclass setting is decomposed into L one-versus-all problems: one binary problem for each class. L calibrators are derived, each one independently calibrating the probability of one class. One problem of this approach is the lack of calibration data for each of the L problems for many classes (if we take 25000 ImageNet samples for calibration, each of the 1000 binary calibration problems has only 25 images available). Another issue is that because each component of the probability vector is changed independently, the model prediction may change."
>
> > What is a reference for "I-Max binning".. I believe it is first mentioned on page 7 (from my searching the paper..), and it scores very well.. (Table 1, on Imagenet dataset/models) Also: why include it if the probabilities can sum to more than 1.0 with this method ? (for some evaluation scores such as log loss, perhaps for ECE too, this could lead to cheating by a method...)
>
> The paper was cited in the related work, but the reference was missing after the method's introduction. It's corrected. We still include this method because we only look at the confidence value (not the full probability vector). We highlighted this limitation because it is important to know for potential users of this method, and it is not mentioned in the original paper. We think that in our paper, no cheating occurs (again, because we only consider the confidence value), but you are right that cheating might occur for metrics using the full vector (like classwise-ECE).
>
> > the authors use 'confidence' (eg on page 4 when they say 'beyond just considering confidence'), but they define it in passing on page 3 as 'the confidence is the top class probability' (top probability as opposed to the probability assigned to other, non-top, classes)... promote it or highlight this technical definition better [...]
>
> In the updated paper, we promoted this definition in the abstract, introduction, and problem setting. It should be clearer now.
>
> > other clarity comments
>
> We considered all of these comments in the updated paper (except the proposition of renaming into top-versus-rest).

---

### Author Response · Authors · 2023-11-16
**General response to reviewers**

We would like to thank all the reviewers for spending time reading, understanding, and reviewing our paper.

The main strengths (reported by several reviewers) of the paper are:
* the idea is simple (reviewers JkUP, fZjD, and 75D8)
* the experiments are comprehensive and show good results (reviewers JkUP, fZjD, TXd7, and 75D8)
* the idea is interesting from a practical point of view (simple, efficient, and model-agnostic) (reviewers JkUP, and fZjD)
* the paper is fairly clear (reviewers JkUP, and TXd7)

The main weaknesses (reported by several reviewers) are:
* the paper's clarity can be improved, as well as the structure of related work and problem-setting (reviewers JkUP, fZjD, and TXd7)
* there is no theoretical study, and the discussion of the approach and results is limited (reviewers fZjD and TXd7)

We thank the reviewers for reporting the strengths of our work. To address the weaknesses and questions, we updated our paper. We restructured sections 2, 3, and 4 and merged them into two sections as advised. We also clarified some points in the results and the approach description. We hope that these changes make the paper clearer.

We also want to highlight the reproducibility and practical use of our work. We use public datasets and pre-trained models. We released the code (on Anonymous Github for the moment) to reproduce our figures and some lines of the tables. We plan to release later the complete code to reproduce all the experiments. Even without our code, our approach can easily be adapted to existing calibration methods by changing just a few lines of code.

---

> ### Author Response · Authors · 2023-11-22
> **One day left**
>
> Dear reviewers JkUP, TXd7, and 75D8, we would like to remind you that the discussion period ends in less than a day. We have made significant efforts to provide detailed responses to your reviews and updated our paper following your recommendations. We clarified the contributions of our work and answered your questions. We are enthusiastic about discussing with each of you and sincerely hope you will take the time to respond.

---

### Meta-Review · Area_Chair_hTPa · 2023-12-11

**Metareview:**

In this paper the authors introduce a new method for the calibration of trained neural networks for classification problems in which the cardinality of the prediction set is quite large. The authors provide a simple and practical technique by posing the problem as one binary calibration. The authors demonstrate the efficacy of their work on image classification problems of ImageNet, CIFAR-10 and CIFAR-100. The reviewers commented positively on the problem addressed by the authors, and a solution that is well-motivated and simple to implement. The reviewers also commented negatively on the quality of the presentation, the lack of theoretical justification, and the lack of comparison with other state-of-the-art methods, and the overall premise of the approach (i.e. just calibrating the top prediction).

After the rebuttals, two reviewers leaned slightly in favor of acceptance but no reviewer argued as a champion for accepting this paper. Given the lack of a champion, I took the opportunity to see if I could provide that endorsement. After my reading of the paper, I agree with the strong concern that the lack of a theoretical justification for the method (or guarantees on the performance of the method) was concerning. Additionally, the limited application of focusing on the top prediction felt like a limitation of the applicability of the method. For these reasons, I can not act as champion and the paper will not be accepted to this conference. The authors are encouraged to clean up the presentation and consider testing their method on other problems whether the cardinality is vastly larger, such as language generation.

**Justification For Why Not Higher Score:**

Heuristic method. Limited scope of method.

**Justification For Why Not Lower Score:**

N/A

---

### Decision · Program_Chairs · 2024-01-16

Reject